# Do Bayesian Neural Networks Actually Behave Like Bayesian Models?

Gábor Pituk [1]  Vik Shirvaikar [1]  Tom Rainforth [1]

## Abstract

We empirically investigate how well popular approximate inference algorithms for Bayesian Neural Networks (BNNs) respect the theoretical properties of Bayesian belief updating. We find strong evidence on synthetic regression and real-world image classification tasks that common BNN algorithms such as variational inference, Laplace approximation, SWAG, and SGLD fail to update in a consistent manner, forget about old data under sequential updates, and violate the predictive coherence properties that would be expected of Bayesian methods. These observed behaviors imply that care should be taken when treating BNNs as true Bayesian models, particularly when using them beyond static prediction settings, such as for active, continual, or transfer learning.

## 1. Introduction

Bayesian neural networks (BNNs) extend the principles of Bayesian inference (Robins & Wasserman, 2000; Jaynes, 2003) to neural networks by placing prior distributions on the model parameters and updating them based on observed data (MacKay, 1992b; Hinton & Van Camp, 1993; Neal, 2012). This approach promises benefits such as principled uncertainty quantification (Neal et al., 2011; Kendall & Gal, 2017), robustness to distribution shift (Daxberger et al., 2021b), and coherent belief propagation (Gal et al., 2017), often cited as important for tasks such as out-of-distribution detection, sequential decision-making, and active learning.

Unfortunately, the practical implementation of Bayesian inference in neural networks is fundamentally intractable, necessitating the use of approximate inference methods. Despite typically being crude approximations of the highly complex BNN posterior in parameter space, these methods generally offer better predictive performance and uncertainty calibration than deterministic networks in multiple

settings (Izmailov et al., 2021b). However, due to the approximate nature of this inference, it is no longer a given that they will still conform to Bayesian ideals.

Why does it matter if BNNs do not behave like Bayesian models? Firstly, several recent efforts seek to improve BNNs by refining priors (Fortuin, 2022) or proposing methodological or algorithmic improvements for enhancing the fidelity of approximate inference (Papamarkou et al., 2024). However, if BNNs fail to respect Bayesian principles such as consistent prior updating and meaningful posterior uncertainty, then we cannot trust that objectively better priors or more faithful inference will translate to performance gains. This is exacerbated by the fact that it is challenging to even precisely evaluate the accuracy of inference in high-dimensional BNNs, so most research has subsequently focused only on improving final predictive performance and calibration metrics of the model-averaged solution (Guo et al., 2017; Wilson & Izmailov, 2020).

Secondly, the adherence of BNNs to Bayesian principles is of direct importance in many contexts. For example, popular active learning acquisition functions such as BALD (Houlsby et al., 2011) and EPIG (Bickford Smith et al., 2023) assume the model will be updated in a true Bayesian manner, and so will become misaligned if this fails to hold. Meanwhile, in settings such as continual and transfer learning, it is essential that sequential updates do not forget about old data (Shwartz-Ziv et al., 2022; Wang et al., 2024). More generally, the conformity of BNNs to Bayesian ideals is important whenever we go beyond static prediction settings, such as when we wish to capture reducible (epistemic) uncertainty, make joint predictions, reason about possible future data, or sequentially update our model.

In this paper we critically evaluate the behavioral properties of popular BNN algorithms. Whereas previous studies have focused on static predictive performance and inference convergence diagnostics (Izmailov et al., 2021b; Sommer et al., 2024), our study aims to more systematically investigate the adherence of BNNs to the expected behavior of true Bayesian models. We find on synthetic regression tasks and the CIFAR and IMDB image and text classification settings of Izmailov et al. (2021b) that BNNs fail to preserve key features of Bayesian inference. In particular, we show that, for the tasks considered, (1) approximate

---

[1]Department of Statistics, University of Oxford, Oxford, UK. Correspondence to: Gábor Pituk <pitukgabor@gmail.com>.

*Proceedings of the 42nd International Conference on Machine Learning*, Vancouver, Canada. PMLR 267, 2025. Copyright 2025 by the author(s).

posteriors lack the functional variability of true Bayesian posteriors (Section 4); (2) sequential inference using BNNs fails to propagate information coherently, leading to significant degradation in accuracy (Section 5.1); and (3) BNN predictive updates are not well-calibrated, failing to retain a self-consistent amount of uncertainty, and thereby violate fundamental information-theoretic properties of Bayesian methods (Sections 5.2 and 6).

Our results, therefore, show that, though BNNs often give powerful static predictive performance, we should be cautious in our expectation that they practically adhere to Bayesian principles. We further provide hope for future improvement, by suggesting that using empirical martingale posterior predictives (Fong et al., 2023) may improve performance over conventional Bayesian model averages.

## 2. Background

Our focus will be on the supervised learning setting as this is where BNNs are most commonly used. Namely, let $\mathcal{D} = \{x_i, y_i\}_{i=1}^n$ be a training set where $x$ denotes the input and $y$ the labeling; our goal is to learn a conditional predictive distribution $Y \mid X = x$.

**BNNs.** To model the data, we use a deep neural network $f_\theta(x)$ with model parameters $\theta$. Instead of training a single empirical risk minimizer $\hat{\theta}_{\text{SGD}}$, BNNs elicit a prior $\pi(\theta)$ over model parameters, condition on $\mathcal{D}$ observed through an i.i.d. observation model $Y_i \mid x_i \sim p(y_i \mid f_\theta(x_i))$, and approximate the posterior distribution $\pi(\theta \mid \mathcal{D}) \propto \pi(\theta) \prod_{i=1}^n p(y_i \mid x_i, \theta)$ (Savage, 1972). The posterior predictive $p(y^* \mid x^*, \mathcal{D}) = \int p(y^* \mid x^*, \theta)\pi(d\theta \mid \mathcal{D})$ can then be used to predict on new data by marginalizing over the posterior, producing the Bayesian model average (BMA, Section A), which is the optimal predictive distribution from a decision-theoretic perspective if the posterior truly represents our beliefs (Fragoso et al., 2018).

**Approximate inference.** The intractability of Bayesian inference in deep learning means that practitioners resort to approximate inference algorithms. Popular practical approaches include localized approximations around a single mode such as variational inference (VI) (Graves, 2011; Blei et al., 2017), Laplace approximations (MacKay, 1992a; Daxberger et al., 2021a), and stochastic weight averaging (SWAG) (Maddox et al., 2019), as well as more expressive but computationally more expensive sampling algorithms (Welling & Teh, 2011; Wenzel et al., 2020). Full-batch Hamiltonian Monte Carlo (HMC) (Neal et al., 2011) is typically considered state-of-the-art; however, it is computationally infeasible to run even for smaller networks.

**Benefits of BNNs.** The Bayesian deep learning literature highlights various attractive theoretical properties of Bayesian inference (Blundell et al., 2015; Wilson, 2020;

Izmailov et al., 2021b; Papamarkou et al., 2024). The maximum a posteriori (MAP) solution has been observed to be overconfident (Guo et al., 2017), which the BMA improves on (Kendall & Gal, 2017; Izmailov et al., 2021b). This is achieved by explicitly accounting for multiple hypotheses, where the ensemble of multiple viable models produces principled uncertainty estimates and prevents overfitting. In practice, BNNs have seen measurable empirical success in predictive tasks, outperforming the MAP estimate in multiple settings (Izmailov et al., 2021b).

## 3. Related Work

BNNs have faced critique. For example, Bayes can be suboptimal in misspecified models (Wang & Blei (2019), Section K.2). BNNs are also often criticized for using uninformative priors that fail to reflect relevant beliefs (Tran et al., 2022; Fortuin, 2022). Although prior engineering is an active research area, objective priors that place mass on non-meaningful predictive representations are widely used (Izmailov et al., 2021b; Papamarkou et al., 2024). In addition, the fidelity of approximate inference in BNNs is routinely questioned, with theoretical pathologies noted by Foong et al. (2020) and Coker et al. (2022). Approximate sampling cannot faithfully capture highly complex BNN posteriors (Wiese et al., 2023), and many schemes make inappropriate assumptions resulting in a crude posterior approximation. These issues inevitably lead to downstream practical issues, e.g. BNNs have been shown to underperform under covariate shift (Izmailov et al., 2021a; Sharma et al., 2023).

Knoblauch et al. (2022) highlight that these key assumptions—correct model specification, meaningful priors, and computationally feasible updates—underpin the theoretical guarantees of Bayesian inference, such as consistency and asymptotic optimality (Bernardo & Smith, 2009), but are fragile in practice, particularly for BNNs. As a result, several *generalized Bayes* approaches have emerged in the wider literature to retain the useful properties of Bayesian inference without strict adherence to full Bayesian updating. Two notable examples are Gibbs posteriors (Bissiri et al., 2016), which replace the log-likelihood with a general loss function, and PAC-Bayes (McAllester, 1999), which balances empirical fit and model complexity.

In BNNs, a generalized Bayesian approach of particular relevance is *cold posteriors* or power posteriors, where the posterior is artificially sharpened to improve BMA performance across different settings (Grünwald, 2012). Wenzel et al. (2020) consider the family of distributions

$$\pi_T(\theta; \mathcal{D}) \propto \left(\pi(\theta) \prod_{i=1}^n p(y_i \mid x_i, \theta)\right)^{1/T}$$

for "temperatures" $T < 1$, and demonstrate that approximate tempered posteriors outperform the Bayes posterior

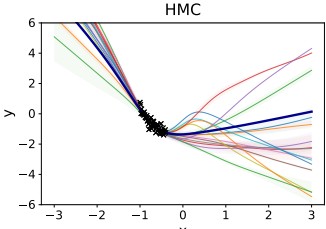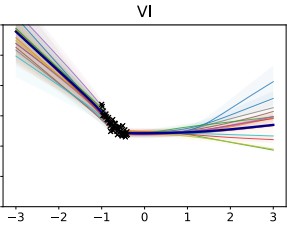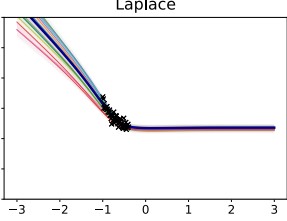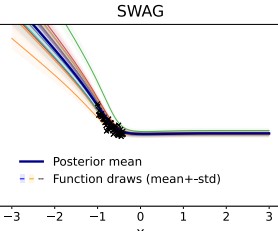

Figure 1: **Synthetic regression posterior functional variability.** Functional samples are drawn from approximate posteriors on the synthetic regression task. The posterior means are similar for all four algorithms. The HMC approximate posterior puts weight on more complex and qualitatively more varied functions compared to the other approximate posteriors, especially in the region $x > 0$, but does not seem to account for the potential of non-monotonic behavior in the region $x < -1$. Despite having somewhat similar point uncertainties, the VI approximation represents less complex and less varied range of functions. Laplace and SWAG fail to capture meaningful uncertainty or functional complexity, especially for $x > 0$.

at $T = 1$. This suggests approximate samplers may struggle to explore enough qualitatively different modes of the posterior, hinting at problems with the model setup. Noci et al. (2021) and Izmailov et al. (2021b) point to data augmentation, curation, and prior issues, but no single cause has been identified (Nabarro et al., 2022). Others argue that posterior tempering is not necessarily against Bayesian principles (Kapoor et al., 2022; Zhang et al., 2024).

Elsewhere, Sharma et al. (2023) showed that fully stochastic BNNs do not typically outperform their *partially stochastic* counterparts. They argue for a holistic view of BNNs as probabilistic algorithms with good predictive performance rather than true Bayesian models. Izmailov et al. (2021b) systematically evaluate the predictive performance of approximate inference against a hugely expensive HMC baseline. While HMC fails to mix in weight space, they argue it mixes well in predictive space, and practical BNN approximate posteriors make similar predictions to HMC. However, Sharma et al. (2023) note that their HMC baseline still exhibits correlations in within-chain predictions, undermining claims that its samples represent the true posterior.

In terms of applications, Bickford Smith et al. (2024a;b) raise issues with BNNs in active learning, while Kirsch et al. (2022) argue for the applicability of Bayesian deep learning joint predictives in active learning and sampling, but find that approximate BNNs fall short in downstream predictive performance. Meanwhile, in continual learning, BNNs have been shown to exhibit *catastrophic forgetting*, where learning a new task worsens performance on previous ones (Chen et al., 2021; Kessler et al., 2023; Wang et al., 2024).

Alternatives to BNNs often aim to capture uncertainty directly in predictive space. For example, epistemic neural networks directly learn a joint distribution over predictions (Osband et al., 2023), while evidential deep learning learns the parameters of a higher-order distribution (Sensoy et al., 2018). Mlodozeniec et al. (2024) assess the "implicit Bayesianness" of predictive updates via *exchangeability*

with observed data. They find through repeated re-training on small-scale tasks that non-Bayesian algorithms may better satisfy this criterion than Bayesian approximations, suggesting that Bayesian deep learning uncertainty can be quantified without explicit priors or posterior updates.

We now shift the focus away from static predictive performance and investigate how practical BNNs behave with respect to the theoretical properties of Bayesian models.

## 4. What Do BNN Functional Posterior Predictives Look Like?

To start our investigation into the adherence of BNNs to Bayesian ideals, we consider BNNs applied to synthetic regression tasks, and measure empirically to what extent these methods behave like true Bayesian models. We deliberately focus here on regression instead of classification as it more clearly highlights nuances of the approximate posterior predictive, noting that we can always consider regressing to class probabilities as a special case.

We use two fully connected BNN architectures with hidden layers of size 32, 32, 16, and 128, 256, 128, 64, respectively. To capture heteroskedastic noise, we choose the observation model $Y \mid x, \theta \sim \mathcal{N}(\mu_\theta(x), \sigma_\theta^2(x))$ where $f_\theta(x) = (\mu_\theta(x), \sigma_\theta^2(x))$, but Section I shows that our findings also generalize to homoskedastic observation models. For training, we consider the mean-field variational inference (MFVI) (Blei et al., 2017), (diagonal) Laplace (Daxberger et al., 2021a), and SWAG algorithms (Maddox et al., 2019), as well as the more computationally intensive Hamiltonian Monte Carlo (HMC) sampler (Neal et al., 2011), generally considered state of the art. Section B gives background on the algorithms, Section G details the experimental setup, Section H shows the desired ground-truth behavior of analytic Bayesian linear regression inference, and Section J assesses the robustness of our findings across network size, input dimension through feature expansion, and hyper-parameterization.

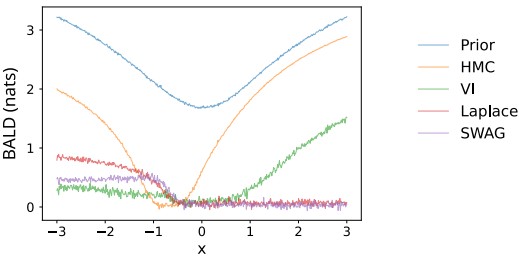

Figure 2: **BALD scores for approximate posteriors shown in Figure 1.** The HMC approximate posterior uncertainty appears sensible in this instance, increasing in regions without observations. Others attribute less importance to samples far from existing observations, and more importance to observing further data around $x = -1$.

### 4.1. What is the Posterior Function Space Variability of Different Approximate BNN Posteriors?

We begin by questioning whether the different approximate posteriors can capture epistemic uncertainty by putting weight on all the models likely to have generated the observed data. BNNs can be described as distributions over the network parameters $\theta$, which we refer to as the *weight-space* or parameter-space characterization. Equally, they can be considered as distributions over predictive functions, $x \mapsto (\mu(x), \sigma(x)))$ in the case of our regression setup, by pushing forward the weight-space distribution through the network $f_\theta(x)$ at any $x$, the so-called *function-space* or predictive-space characterization (Izmailov et al., 2021b). Going beyond the posterior mean (BMA) $\mathbb{E}_{\theta \sim \pi(\theta \mid \mathcal{D})} f_\theta(x^*)$ and point uncertainties $p(y^* \mid x^*, \mathcal{D})$ and $p(\mu(f_\theta(x^*)) \mid x^*, \theta)$, we are interested in the functional properties of the approximate BNN posteriors in predictive space.

BNN use cases relying on the full posterior (some of which will be detailed in Section 5.1) require that the posterior approximations behave reasonably in function space to represent a range of likely models to describe the data. In particular, they expect function draws from the posterior to preserve the characteristics of the prior predictive in regions without observations. We investigate if this holds for practical BNNs in our regression setting.

Figure 1 shows function draws from different approximate posteriors trained on a synthetic regression task. Based on the samples, HMC retains more of the prior function complexity in the high-uncertainty region $x > 0$ compared to the other schemes which overly favor simpler functions. However, even HMC does not account for qualitatively distinct function behaviors in the region $x < -1$.

Variational inference (VI), when appropriately tuned (see Section B.2), seems to produce somewhat sensible point uncertainty, increasing in regions without observations. How-

ever, the function draws are more similar to each other than the HMC approximate posterior. Note that VI's behavior is sensitive to a range of parameters, including the choice of $\beta$ in Equation (16): smaller $\beta$, smaller initial variances, and shorter training time all trade off a closer fit against regularization towards the prior.

The Laplace and SWAG function-space posteriors meanwhile collapse to very specific MAP-like behaviors. Their function draws appear to be mostly simple parallel transformations of their posterior mean; they fail to even produce increasing point uncertainty in the region $x > 0$. Both algorithms make local Gaussian approximations around a MAP estimate, averaging models in this region of the parameter space. In this instance, this local averaging does not seem to capture qualitatively different models to describe the data.

### 4.2. How Much Uncertainty Reduction Should We Expect from Observing Further Data Points?

To gain further insights into Bayesian model variability and connect epistemic uncertainty with point uncertainty, we now consider the information-theoretic BALD score (Houlsby et al., 2011), often used in Bayesian active learning as an acquisition function. At an input $x^*$, it is defined as the mutual information between $\theta$ and $Y^* \mid x^*$, or equivalently, the expected reduction in weight-space uncertainty when observing a data point at $x^*$.

Its values for different approximate BNN posteriors on our regression task are shown in Figure 2. The HMC approximate posterior attributes more importance to observing data far from existing observations, whereas other approximate posteriors would be more informed by further observations around $x = -1$, which may be undesirable in learning systems. Laplace and SWAG appear so confident in the generalizing behavior for $x > 0$ that they expect negligible information gain from observing data in this region.

Despite the differing posterior functional properties and generalization behavior and shortcomings uncovered, the BMAs of these approximate posteriors are all similar and in line with our expectations. This observation motivates our investigation into function-space behavior of practical BNNs beyond model-averaged static performance.

## 5. Is Data Captured in Intermediate Posteriors Weighted Consistently with New Data?

In this section we demonstrate deviations of practical BNNs from Bayesian behavior in settings requiring sequential decision making in between observing data, utilizing epistemic uncertainty directly and using intermediary posteriors in subsequent steps of inference. Our tests will be particularly relevant to the setting of active, continual, and transfer learning as discussed below.

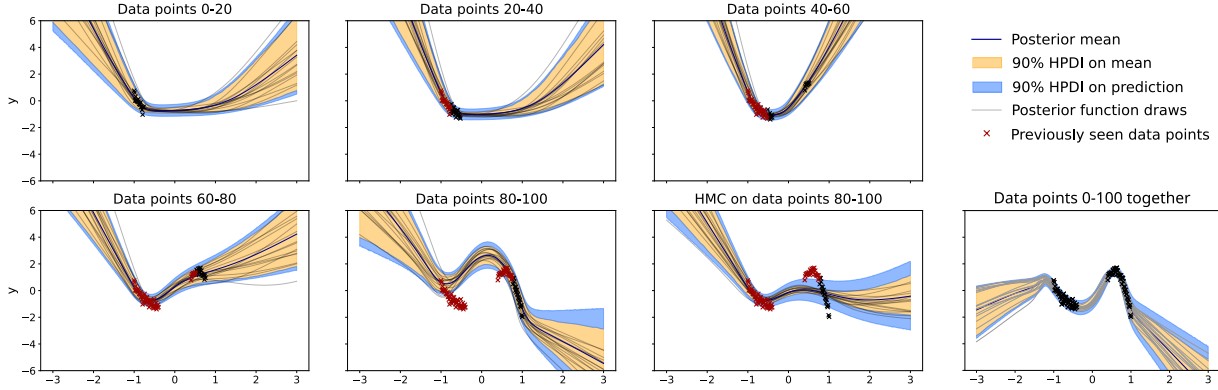

**Figure 3:** **Sequential updating experiment with VI on larger network.** Each step's prior is the posterior approximation from the previous step. Point uncertainty is shown through high probability density intervals (HPDI) on the predictive mean and the observation. Data points currently trained on are shown in black, previous ones in red. The last step of inference is repeated with HMC, as well as a single-step VI approximation is shown for reference. *Catastrophic forgetting* occurs: fitting the new data leads to worse fit on earlier data, and observed information is gradually lost.

*Active learning* and *experimental design* (Atlas et al., 1989; Settles, 2009; Rainforth et al., 2024) are sequential learning paradigms, where popular acquisition functions such as BALD (Houlsby et al., 2011) or EPIG (Bickford Smith et al., 2023) are based on Bayesian model uncertainty. We might model the data with a BNN (Houlsby et al., 2011; Gal et al., 2017) and sequentially update beliefs with the fresh observations. This, however, requires that BNNs respect the sequential properties of Bayesian updating.

*Continual learning* focuses on an agent sequentially learning a number of tasks; Bayesian formulations using online Laplace approximations, VI, or function-space VI have been proposed (Wang et al., 2024, see Section 3.1), where it is crucial for the learner not to forget about old observations.

In *transfer learning*, a learner captures the knowledge of a pretraining task to improve performance on a downstream task. There have been Bayesian models of transfer learning (Shwartz-Ziv et al., 2022; Suder et al., 2023) that propose using an intermediary SWAG posterior learned on the pretraining task. Here, there is an implicit assumption that this approximate posterior is able to functionally capture pretraining knowledge to inform training on the downstream task. We investigate empirically if such applications of BNNs are justified from a Bayesian modeling perspective.

### 5.1. Are Sequential Belief Updates Coherent?

An important aspect of Bayes' rule is the following invariance with respect to the order of observations.

**Proposition 5.1** (Sequential Coherence of Bayesian Inference, Bernardo & Smith (2009)). *If $\mathcal{D}$ and $\mathcal{D}'$ are two sets of observations, then the "iterated" posterior $\pi\left((\theta \mid \mathcal{D}) \mid \mathcal{D}'\right)$ is the same as the full posterior $\pi\left(\theta \mid \mathcal{D} \cup \mathcal{D}'\right)$.*

To set up a sequential inference, suppose we have datasets $\{\mathcal{D}^{(i)}\}_{i=1}^N$ which the learner observes sequentially and needs to make decisions in between observing them. Suppose further that we have an initial belief $\pi(\theta)$, and a scheme $\pi(\theta) \mapsto \pi_y(\theta)$ to update beliefs as a function of observed data $y$. We use the incremental beliefs $\left((\pi_{\mathcal{D}^{(1)}}) \cdots\right)_{\mathcal{D}^{(i)}}(\theta)$ to make predictions on a held-out dataset. We then test how these beliefs generated from sequential inference compare to conditioning on all the data in one go, i.e. $\pi_{\{\mathcal{D}^{(i)}\}_{i=1}^N}(\theta)$.[1]

**Synthetic regression.** We partition our synthetic regression dataset into $N = 5$ equal groups based on the $x$ value, and run sequential approximate inference. According to Bayesian coherence, each intermediate posterior should be equal to the single-step posterior trained on all prior data points. We find this not to hold in practice: Figure 3 gradually demonstrates catastrophic forgetting. The final VI approximate posterior differs from the single-step approximate posterior, not fully respecting the first half of the observations. HMC on the last step produces a worse fit than VI, indicating that the intermediary VI posterior after 80 data points may be overconfident. The practically necessary use of $\beta = 0.05 < 1$ masks this inconsistency by overweighting observations 80-100 in the final step of inference. Section J shows a similar or even worse forgetting behavior for Laplace and different network variants.

**Image and text classification.** We now scale this experiment up to real-world model and image and text classification benchmarks. Izmailov et al. (2021b) demonstrated that Bayesian alternatives typically outperform deterministic

---

[1]This experiment requires a closed-form approximate posterior, which holds for VI, Laplace, and SWAG. Although practitioners may retrain on all prior observations each time, we analyze if such algorithms are justified from a Bayesian angle in the first place.

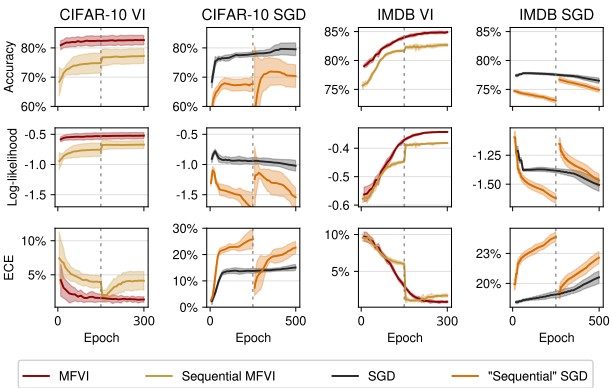

Figure 4: **Sequential MFVI on CIFAR-10 and IMDB.** We show test accuracy, log-likelihood, and expected calibration error (ECE) for the single step and iterated inference (mean and st. dev. across 6 seeds). Under Bayes the two should make the same predictions, but iterated MFVI significantly *underperforms* single-step MFVI across all metrics. VI improves over SGD, and fixes the growing overconfidence of SGD during training. However, in iterated training, the principled Bayesian belief sharing does not improve MFVI more than the heuristic belief sharing improves SGD.

networks in multiple classification and regression settings. We replicate their MFVI experiments on ResNet-20-FRN CIFAR-10, CIFAR-100 (Section F.1), and CNN-LSTM IMDB, taking their implementation and hyperparameters.

We randomly split the training sets into two splits $\mathcal{D}^{(1)}$ and $\mathcal{D}^{(2)}$. We train an intermediary MFVI approximate posterior $\pi_{\mathcal{D}^{(1)}}(\theta)$ on the first split, and pass this as a prior to infer a second MFVI approximate posterior $\left(\pi_{\mathcal{D}^{(1)}}\right)_{\mathcal{D}^{(2)}}(\theta)$ on the second split. As the number of sequential inference steps grows, we expect estimation error to increase and final approximation quality to deteriorate. To focus on the

intrinsic algorithmic difference instead of estimation error accumulation, in this experiment we take $N = 2$ random splits and do only a single step of retraining.

Figure 4 compares the iterated posterior against the single-step posterior in terms of the predictive performance metrics of the BMA. Contrary to Bayesian expectations, iterated VI performs significantly and consistently worse than single-step VI, even though $\left(\pi_{\mathcal{D}^{(1)}}\right)_{\mathcal{D}^{(2)}}(\theta)$ is still an improvement over $\pi_{\mathcal{D}^{(1)}}(\theta)$. To evaluate this improvement, we run a *"sequential SGD"* baseline where an SGD solution is trained on $\mathcal{D}^{(2)}$ using the first-split SGD solution $\arg\max_\theta \pi(\theta \mid \mathcal{D}^{(1)})$ as an initializer without any additional training guidance (such that the only influence of $\mathcal{D}^{(1)}$ is from the initialization). This deliberately non-Bayesian bias to share beliefs leads to a similar improvement from the intermediate MAP to the "sequential" biased MAP as the improvement from $\pi_{\mathcal{D}^{(1)}}(\theta)$ to $\left(\pi_{\mathcal{D}^{(1)}}\right)_{\mathcal{D}^{(2)}}(\theta)$. Based on this experiment, we do not find the use of VI to be justified in sequential inference contexts by its Bayesian motivations alone.

### 5.2. How Strong are the Beliefs Captured in an Intermediary Posterior?

We now turn back to synthetic regression and test how strong the knowledge captured in the intermediary posterior is compared to subsequent observations. To precisely quantify the amount of information encoded in an intermediary posterior and fresh observations, we train an approximate posterior on 50 data points generated from $Y_i \mid x_i \sim \mathcal{N}(0.5, 0.05^2)$. The learner then gradually observes data from the shifted distribution $Y_i \mid x_i \sim \mathcal{N}(-0.5, 0.05^2)$ with this approximate posterior used as the prior. Training on $n$ new data points should give a predictive mean of $\frac{50-n}{50+n} \cdot 0.5$ shrunk by the original prior towards zero. This means we should need to observe more than 50 new points to shift the posterior predictive mean negative if points are being weighted equally.

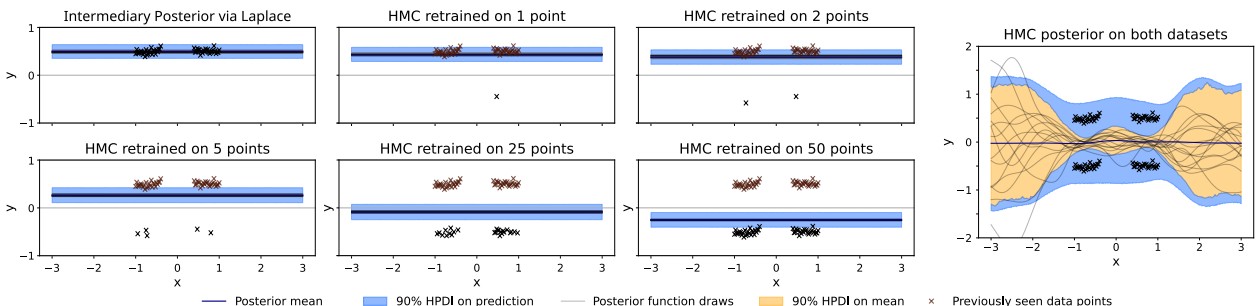

Figure 5: **Intermediary Laplace posterior experiment.** An intermediary posterior is learned from the original dataset, then subsequent HMC inference is performed with this intermediate posterior as prior, observing data from a shifted distribution. We show the larger network with feature expansion, but the result generalizes; see Section J. After only 25 points, the posterior mean becomes negative: the new observations appear to be weighted around twice as much as the pretraining observations, despite the shrinkage of the intermediary Laplace approximate posterior towards the MAP estimate. Single-step HMC inference on the original and shifted datasets at the same time is able to produce a sensible posterior mean.

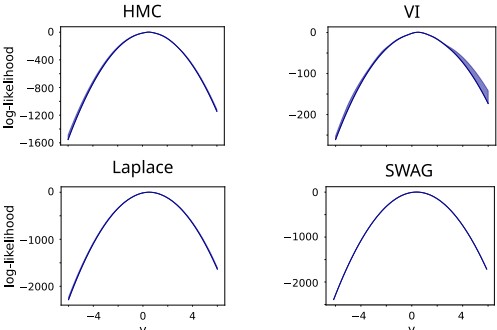

Figure 6: **Cross-sectional predictive log-likelihoods.** We compute log-likelihoods at $x = 0$ on the pretraining task observing data centered at $y = 0.5$. The tail behavior of different posterior predictives exhibit significant differences. In particular, VI is significantly heavier-tailed than others despite over-weighting observations due to taking $\beta < 1$. Laplace and SWAG predictives resemble Gaussian tails, and are lighter tailed than the HMC predictive.

We use Laplace approximation to approximate the intermediary posterior, which we then pass to HMC inference as a prior, observing points from the shifted distribution. Figure 5 shows that *catastrophic forgetting* is observed: recent observations are weighted more heavily compared to earlier observations (Wang et al., 2024). Either the Laplace approximation has failed to properly incorporate the provided data, or HMC has failed to mix, potentially stuck in an ill-conditioned optimum, or some combination of the two.

For VI, we find even stronger forgetting behavior, underpinned by the less confident VI intermediary posterior predictive; see Section J.4. Neither of these approximate inference schemes appears to faithfully encode pretraining data in its posteriors for use in downstream tasks, nor to capture functional knowledge from pretraining.

We further examine the tail behavior of the posterior predictive distributions in Figure 6, as this can shed light on how flexible the current model is to change with future observations. We see that the Laplace predictive is lighter-tailed than HMC; it is more confident in the pretraining observations. Given that we still observe forgetting when using these lighter-tailed Laplace approximations as priors, this suggests inaccuracies in the HMC inference.

# 6. Evaluating the Predictive Coherence of Belief Updates with Martingales

In the previous section, we examined whether coherence in predictive beliefs is maintained when sequentially updating the parametric posterior using approximate Bayesian inference with increasing amounts of externally provided data. We now instead investigate whether the distribution of predictions made by BNNs is itself self-consistent with how

the model itself updates when those predictions are fed back into the training as pseudo-data. This tests BNNs against the important property of Bayesian models that they should not hallucinate new information without external data or destroy information from previous data.

In particular, given an observed dataset, any predictive model somehow manipulates the information contained in the data to produce a predictive distribution (Dawid, 1984). Suppose we then generate pseudo-data from our model, impute it into our training set, and perform another update. If the model updates are well-calibrated, then on average, across all possible imputations, the new predictive distribution should not change, as no new information has been added. If it does, it implies that information from the initial observed data has been lost or transformed through our updating scheme, resulting in miscalibration or "hallucination" in predictive space (Falck et al., 2024).

A diagnostic based on predictive consistency therefore tests whether a model update is retaining a consistent amount of uncertainty with respect to the data-generating process, or if pseudo-information is being introduced through the model's predictions. This diagnostic evaluates posterior model uncertainty not captured by the BMA, thereby assessing the coherence through a more fundamental predictive lens.

How does this relate to Bayesian behavior? The martingale posterior framework of Fong et al. (2023) presents a predictive view of Bayesian inference, where uncertainty arises from missing data, not unknown parameters. Under this view, the Bayesian task is not to update a prior on parameters, but to assign a predictive distribution to missing data conditional on observed data. A result from Doob (1949) shows that this predictive uncertainty is equivalent to parametric uncertainty: incoherence in one implies incoherence in the other. While this is usually framed parametrically—an incorrect update leads to incorrect predictions—we focus on the mirrored view: if predictions are miscalibrated, the parametric update must also be flawed.

We now provide a brief technical exposition on the martingale posterior framework, with further details in Section C. Let $\Theta \sim \pi(\theta)$ be a parameter of interest with some prior density $\pi(\theta)$ before any data is observed. For observed data $Y_{1:N}$, the posterior mean of this parameter is $\bar{\theta}_N = \mathbb{E}[\Theta \mid Y_{1:N}]$. Note here that we did not need to assume some explicit likelihood form $p(y_{1:N}|\theta)$, we simply need a mechanism to recover $\mathbb{E}[\Theta \mid Y_{1:N}]$ from data. For example, we might simply have a mechanism for training a stochastic neural network given observed data. The following result from Doob (1949) now shows that parametric and predictive uncertainty are equivalent if $\bar{\theta}_N$ is a martingale, that is if its conditional expectation at each step is equal to its previous value, meaning that new data points only propagate uncertainty and do not introduce bias.

**Theorem 6.1** (Martingale Convergence, Doob (1949))**.** *If $\bar{\theta}_N$ is a martingale, with $\mathbb{E}\left[\bar{\theta}_N \mid Y_{1:N-1}\right] = \bar{\theta}_{N-1}$, then under certain measurability and identifiability conditions, $\bar{\theta}_N \to \Theta$ almost surely.*

In other words, when this martingale property holds, rather than viewing belief updates solely as refining our knowledge about $\Theta$, we can interpret them as the resolution of uncertainty about future observations. Each new data point $Y_N$ refines our predictive uncertainty, and as we sequentially impute the unseen $Y_{1:\infty}$, the remaining uncertainty in $\Theta$ vanishes and our sequence of posterior means $\bar{\theta}_N$ converges to a draw from the prior.

Under this approach, Bayesian inference is nothing more than a predictive task, where our goal is to generate different possible imputations of the unobserved data. Uncertainty is then indexed over these hypothetical completions of the dataset. Specifically, suppose we have initial belief $\pi(\theta)$, a belief updating scheme $\pi(\theta) \mapsto \pi_y(\theta)$ that incorporates observed data $y$ into beliefs $\pi$, and a parameterized predictive model $p(y \mid \theta)$. Let $Y^*$ be a random sample drawn from the marginal predictive $p(y^*) = \int p(y^* \mid \theta)\pi(d\theta)$. Then we define the *martingale posterior distribution* (Fong et al., 2023) for this sample as the expected belief update across all possible predictive imputations:

$$MP_{Y^*}\pi(\theta) = \mathbb{E}\left[\pi_{Y^*}(\theta)\right] = \int \pi_{y^*}(\theta)p(dy^*).$$

It can be estimated with the (consistent) *empirical martingale posterior distribution*:

$$\widehat{MP}_{Y^*}^{(N)}\pi(\theta) = \frac{1}{N}\sum_{j=1}^{N} \pi_{y_j^*}(\theta)$$

where $y_j^* \overset{\text{i.i.d.}}{\sim} p(y^*)$ averaging across $N$ trials.

As exact Bayesian updates do satisfy the martingale property, we can now introduce another key property of Bayesian inference relating to its predictive belief updating:

**Proposition 6.1** (Predictive Coherence of Bayesian Inference, based on Fong et al. (2023))**.** *When the belief updating scheme is exact Bayesian inference, the martingale posterior recovers the initial beliefs.*

*Proof.* In the case, for example, when prior and likelihood densities exist and $0 < p(y^*) < \infty$ for $p$-almost every $y^*$,

$$MP_{Y^*}\pi(\theta) = \mathbb{E}[\pi(\theta \mid Y^*)] = \int \pi(\theta \mid y^*)p(dy^*) =$$

$$\int \frac{\pi(\theta)p(y^* \mid \theta)}{p(y^*)}p(dy^*) = \pi(\theta)\int p(dy^* \mid \theta) = \pi(\theta). \square$$

At least in principle, it is possible for this predictive coherence to be satisfied even if our approximate inference scheme is itself inaccurate. Thus one might see "Bayesian"

updating behavior in the predictive model even if our inference is poorly representing the true parameter posterior. However, in our tests below we find BNNs breaking this productive coherence, such that they do not properly maintain information in the data between iterations.

We use the CIFAR-10 dataset for this experiment, and consider taking two random subsets of 4080 images each: a labeled split $(x, y)$, and an unlabeled split $x^*$. We first use the labeled split to train an initial BNN posterior approximation $\pi'(\theta) = \pi_{(x,y)}(\theta)$. For the unlabeled split $x^*$, we use the intermediary $\pi'$-predictive to sample labelings through $\theta \sim \pi'(\cdot)$ followed by $Y^* \mid \theta \sim p(\cdot \mid \theta; x^*)$. We then train a new posterior approximation $\pi''(\theta) = \pi_{(x,y)\cup(x^*,Y^*)}(\theta)$. This label-retrain step is repeated over $N$ trials, allowing us to compute an empirical martingale posterior distribution

$$\widehat{MP}_{(x^*,Y^*)}^{(N)}\pi_{(x,y)}(\theta) = \frac{1}{N}\sum_{j=1}^{N}\pi_{(x,y)\cup(x^*,\hat{y}_j^*)}(\theta)$$

that averages over the $N$ different predictive imputations of the unknown labels $y^*$. We then compare the BMA from the initial approximate posterior $\mathbb{E}_{\theta \sim \pi_{(x,y)}(\cdot)}[f_\theta(x_{\text{test}})]$ and the BMA from the empirical martingale posterior $\mathbb{E}_{\theta \sim \widehat{MP}_{(x^*,Y^*)}\pi_{(x,y)}(\cdot)}[f_\theta(x_{\text{test}})]$. Propositions 5.1 and 6.1 imply that under Bayesian inference and $N \to \infty$ these two model averages should match.

With the goal of analyzing function-space predictive coherency, we apply this uncertainty diagnostic to MFVI and Stochastic Gradient Langevin Dynamics (SGLD) (Welling & Teh (2011), Section B.5). We treat these algorithms as end-to-end belief updating schemes, replicating the setup and their hyperparameters from Izmailov et al. (2021b). Figure 7 shows the result of our martingale posterior experiment on CIFAR-10, comparing the performance of the initial posterior distribution $\pi'$ to that of the martingale posterior. Specifically, we examine Bayesian model average (BMA) metrics of accuracy, log-likelihood, and expected calibration error as projected summaries of posterior predictive performance (see Section A).

Under true Bayesian inference the martingale posterior should recover $\pi'$, and thus their BMA predictions are expected to match. However, Figure 7 shows that the martingale posterior metrics deviate significantly from those of $\pi'$, with deviations exceeding bootstrapped confidence intervals. This divergence indicates that the posteriors obtained from MFVI and SGLD fail to fully capture the correct Bayesian belief updating. Notably, the deviation manifests in improved predictive performance, especially in the case of MFVI. This suggests that the martingale posterior approach of generating ensembles by simulating pseudo-data and retraining may be generally beneficial compared to just using the BMA in BNN setups, if the additional computation cost of doing so can be justified.

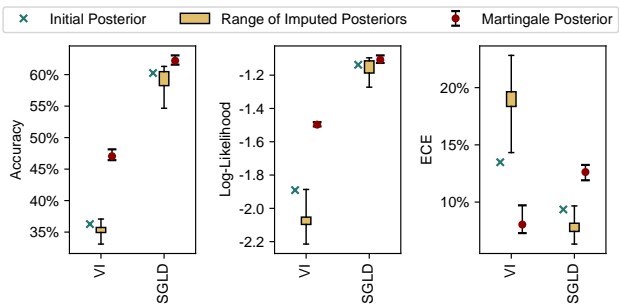

Figure 7: **Martingale posterior BMA on CIFAR-10 subset.** We show the test accuracy, log-likelihood, and ECE for the initial posterior $\pi'$; the full and inter-quartile ranges of imputed posteriors $\pi''$; and the empirical martingale posterior with a 95% bootstrap confidence interval with respect to resampling imputed labelings $\hat{y}^*$. For both VI and SGLD, individual imputed posteriors underperform the initial BMA, but their martingale ensemble outperforms it. Imputing labels should not introduce new information, but here retraining increases performance, falsifying Bayesian coherency.

We further examine the full calibration curves of the model averages in Figure 8. While the confidence of the initial trained posteriors increases on synthetically retrained datasets, consistent with being trained on more data, the martingale posterior exhibits reduced predictive confidence despite improved accuracy. This is evident in the calibration plots: the martingale posterior increases accuracy for a given confidence level on test data. This behavior parallels the well-established benefits of ensembling: averaging over a diverse set of predictive distributions mitigates overconfidence and yields more robust uncertainty quantification.

We have thus used this prediction-powered diagnostic to assess whether BNN updating preserves the correct model uncertainty. In this case, both VI and SGLD fall short, leading to suboptimal ensembling over models. Martingale posterior ensembles provide a partial remedy, performing function-space correction using unlabeled data. Section E.2 strengthens these findings on the IMDB dataset.

## 7. Discussion

We have systematically examined the behavioral properties of common BNN algorithms, highlighting deviations from key characteristics that would be expected of exact Bayesian models. Specifically, we find that approximate posteriors lack functional variability, sequential inference does not propagate prior information coherently, information can be hallucinated or lost when updating, and predictive updates are poorly calibrated.

**Should we avoid BNNs completely?** Definitely not! They have been shown to outperform point estimates and provide effective static uncertainty in multiple settings, which our

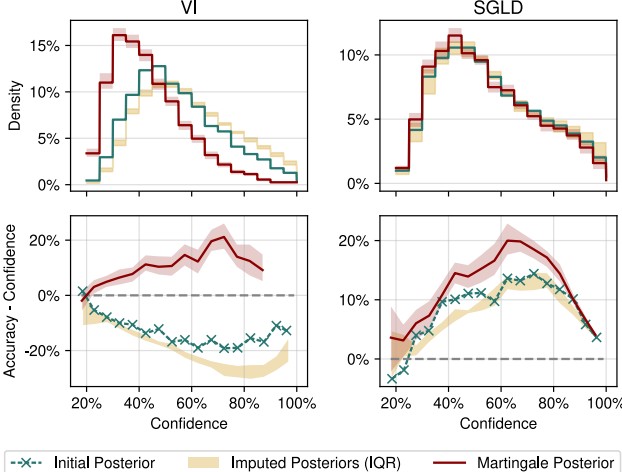

Figure 8: **Predictive uncertainty calibration of the martingale posterior.** Investigating the ECE from Figure 7 further, for the BMA test-set predictions we show the distribution of *confidence*, i.e., maximum predicted class probability **(top)**, and *calibration curves* (Guo et al., 2017), i.e., prediction accuracy minus confidence for test samples falling into a given confidence bucket **(bottom)**. The initial VI posterior is overconfident while SGLD is underconfident. For both, the individual imputed posteriors $\pi''$ produce more confident but not more accurate model-averaged predictions. Their martingale posterior has reduced confidence compared to the initial posterior, leading to under-confident but more accurate predictions. The mismatch is greater for VI.

results do not undermine. We simply argue that practitioners should exercise caution in their expectations and interpretation, especially when applying BNNs in distinctly Bayesian contexts such as active, continual, and transfer learning, where these pathologies may limit their utility. This also applies to empirical evaluation, as good static performance may not translate to other desired behaviors.

Overall, we encourage a *pragmatic perspective*: rather than striving for strict Bayesian adherence, BNNs should be viewed as practical algorithms inspired by Bayesian ideas. This reframing, already acknowledged within the field (e.g. Sharma et al. (2023)) emphasizes the importance of factors such as optimizer design, regularization, ease of training, and explicit solutions for coherency where needed. In particular, although the problems we raise are a direct result of inexact inference, improving inference schemes is not necessarily the solution: accurate inference may be unachievable, and strategies focusing on final behavior could be more effective. Further, entirely non-Bayesian methods might in practice give more coherence in some settings.

Going forward, the improvements seen by the empirical martingale posterior predictives hint at an interesting direction. Though requiring additional computation, these led to gains in predictive performance without any additional data.

## Acknowledgements

The authors gratefully acknowledge Mrinank Sharma for his guidance and insightful comments during the development of this work. VS is supported by the UK EPSRC Centre for Doctoral Training in Modern Statistics and Statistical Machine Learning (grant EP/S023151/1) and Novo Nordisk. TR is supported by the UK EPSRC grant EP/Y037200/1.

## Impact Statement

This paper presents work whose goal is to advance the field of Machine Learning. There are many potential societal consequences of our work, none which we feel must be specifically highlighted here.

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

# A. Bayesian Model Averaging

In this section, we revise and introduce terminology for the Bayesian model average (BMA), and its estimation. We also define the classifier evaluation metrics we use throughout the paper to compare different BMA or point predictions.

## A.1. BMA for Regression and Classification

In the context of Section 2, the decision-theoretically optimal Bayes predictor at fresh input $x^*$ with respect to loss function $l$ can be constructed as the minimizer of the expected posterior loss

$$\hat{y}^*(x^*) = \arg\min_{y^*} \mathbb{E}\left[l(Y^*, y^*) \mid x^*, \mathcal{D}\right] = \arg\min_{y^*} \mathbb{E}_{\theta \sim \pi(\cdot \mid \mathcal{D})} \mathbb{E}\left[l(Y^*, y^*) \mid x^*, \theta\right], \tag{1}$$

see e.g. (Hirshleifer & Riley, 1992). For regression under the $L^2$ loss this corresponds to the posterior mean

$$\hat{y}^*(x^*) = \mathbb{E}_{\theta \sim \pi(\cdot \mid \mathcal{D})} \mu(f_\theta(x^*)) = \int_\Theta \mu(f_\theta(x^*)) \pi(d\theta \mid \mathcal{D}), \tag{2}$$

and for classification under the 0-1 loss this corresponds to the *a-posteriori* most likely class

$$\arg\max_{y^*} p(y^* \mid x^*, \mathcal{D}) = \arg\max_{y^*} \int_\Theta p(y^* \mid x^*, \theta) \pi(d\theta \mid \mathcal{D}). \tag{3}$$

Both predictors rely on marginalization, i.e., posterior-weighted averages over models.

## A.2. Approximating the BMA From Samples

Having access to posterior samples $\theta_1, \ldots, \theta_N \sim \pi(\theta \mid \mathcal{D})$, we can approximate the BMA with the *ensemble* $\pi(\theta \mid \mathcal{D}) \approx \frac{1}{N} \sum_{j=1}^N \delta_{\theta_j}(\theta)$, leading to the following finite mixture posterior predictive approximations:

$$p(\mu(x^*) \mid x^*, \mathcal{D}) = \int_{\theta \in \Theta : \mu_\theta(x^*) = \mu(x^*)} \pi(d\theta \mid \mathcal{D}) \approx \frac{1}{N} \sum_{j=1}^N \delta_{\mu(f_{\theta_j}(x^*))}(\mu(x^*)) \tag{4}$$

for the model mean, and

$$p(y^* \mid x^*, \mathcal{D}) = \int_\Theta p(y^* \mid x^*, \theta) \pi(d\theta \mid \mathcal{D}) \approx \frac{1}{N} \sum_{j=1}^N p(y^* \mid x^*, \theta_j) \tag{5}$$

for the observed $Y^*$. Under some conditions, including $\theta_j$ being independent or samples from an ergodic Markov chain, these likelihood estimators are consistent (Robert et al., 1999; Roberts & Rosenthal, 2004).

This leads to the following (consistent) predictive log-likelihood estimator:

$$\log p(y^* \mid x^*, \mathcal{D}) \approx \log\left(\sum_{j=1}^N p(y^* \mid x^*, \theta_j)\right) - \log N. \tag{6}$$

## A.3. Evaluating BNNs

When evaluating BNNs on a test set, for regression, we use the term BMA to refer to the predictive mean estimates $\hat{y}^*(x_i^*)$; for classification, we refer to the full predictive averaged class probability vector $(p(y^* = k \mid x_i^*, \mathcal{D}))_{k=1}^K \approx J^{-1} \sum_{j=1}^J f_{\hat{\theta}_j}(x_i^*) = J^{-1} \sum_{j=1}^J \mathrm{softmax}(g_{\hat{\theta}_j}(x_i^*))$.

In classification, BNNs are often evaluated along the following metrics. Let $(x^*, y^*)$ be a random sample and its label picked uniformly from the test data distribution.

1. The predictive accuracy is $\mathbb{E}_{(x^*, y^*)} \mathbf{1}\{y^* = \arg\max_y p(y \mid x^*, \mathcal{D})\} = N_{\text{test}}^{-1} \sum_{i=1}^N \mathbf{1}\{y_i^* = \arg\max_y p(y \mid x_i^*, \mathcal{D})\}$;

2. (normalized) log-likelihood is $\mathbb{E}_{(x^*,y^*)} \log p(y^* \mid x^*, \mathcal{D}) = N_{\text{test}}^{-1} \sum_{i=1}^{N} \log p(y_i^* \mid x_i^*, \mathcal{D})$;

3. and finally, to introduce calibration and expected calibration error (ECE) as defined by Guo et al. (2017), let us define

  - *confidence* as the maximum predicted class probability $C(x^*) = \max_y \text{softmax}(g_\theta(x^*))_y$;
  - *calibration error* as absolute deviation of calibration from confidence at a given confidence level

$$Calibr(x^*) = \left| \mathbb{E}_{(x^*,y^*)} \left[ \mathbf{1}\{y^* = \arg\max_y p(y \mid x^*, \mathcal{D})\} \,\middle|\, C(x^*) \right] - C(x^*) \right|;$$

  - and the *expected calibration error* as $\mathbb{E}_{x^*} Calibr(x^*)$.

The lower the ECE, the more "reliable" the output softmax probabilities are; the SGD solution has been often found to be overconfident in its predictive softmax probabilities (Guo et al., 2017; Izmailov et al., 2021b).

## B. Approximate Inference Algorithms

In this section we introduce some of the popular approximate Bayesian inference algorithms for BNNs.

### B.1. Hamiltonian Monte Carlo

Hamiltonian Monte Carlo (HMC) (Neal et al., 2011; Betancourt, 2017) is a Markov-chain Monte Carlo (MCMC) algorithm that is the state of the art for sampling from BNN posteriors (Wenzel et al., 2020; Izmailov et al., 2021b). In high-dimensional distributions, such as BNN posteriors, there are many counter-intuitive phenomena not found in low dimensions, such as the concentration of mass to a thin shell some distance away from the mean. An effective sampler needs to explore this *typical set* of points of high density efficiently. Classical algorithms such as random-walk Metropolis or Gibbs are not suitable for this: because the shell is thin, random-walk proposals are rejected or stay local, and Gibbs does not mix well because the level sets of the density are non-convex.

HMC operates on the state space extended by the momentum vector of a particle whose movement is governed by Hamiltonian dynamics. The proposal is computed by numerically integrating the path of the particle, using first-order gradient information. This dynamics allows the particle to travel along the typical set so that the proposal is far away, but still has a high probability of acceptance.

More concretely, assume we wish to sample from $\pi(q)$. We extend the target by a momentum vector $p$ of the same dimension as $q$: $\pi(p,q) = \pi(q)\pi(p \mid q)$ for some suitable distribution $\pi(p \mid q)$. Let us define $H(p,q) = -\log \pi(p \mid q) - \log \pi(q) = K(p,q) + V(q)$ to be the Hamiltonian, $K(p,q)$ the kinetic energy, and $V(q) = -\log \pi(q)$ the potential energy of a particle.

Motivated by Hamiltonian dynamics, we consider simulating the state of a particle over time $t$ governed by the differential equation

$$\frac{\mathrm{d}q}{\mathrm{d}t} = \frac{\partial H}{\partial p} = \frac{\partial K}{\partial p} \tag{7}$$

$$\frac{\mathrm{d}p}{\mathrm{d}t} = -\frac{\partial H}{\partial q} = -\frac{\partial K}{\partial q} - \frac{\partial V}{\partial q} \tag{8}$$

for some period to get $p^*$ and $q^*$. By change of variables, it is easy to check that $H$ stays constant, so $\pi(p,q) = \pi(p^*,q^*)$. Thus, if $(p,q)$ is a point in the typical set, so is $(p^*,q^*)$. Note that $\frac{\partial V}{\partial q}$ can be computed from an unnormalized density $\bar{\pi}(q)$, which is crucial for sampling from the Bayesian posterior where the normalizing constant is intractable.

To build a MCMC algorithm based on this idea we construct a random-walk Metropolis proposal by numerically simulating the dynamics using step size $\epsilon$ for $L$ iterations, with additional tricks to ensure irreducibility and symmetry of the proposal distribution. Note that under perfect simulation the densities are equal, hence the acceptance probability is always one. The accept-reject step corrects for the integration error. Marginally, the $q$ state of the particle will efficiently explore the typical set of the target distribution.

The No-U-Turn sampler (NUTS) (Hoffman et al., 2014) is an adaptive variant of HMC, adjusting the step size $\epsilon$ and number of steps $L$, with the goal of maximizing the distance between the current point and the proposal without wasting computation. This allows NUTS to discover the high-dimensional state space faster and avoid the need to tune these hyperparameters manually.

HMC is computationally feasible for sampling from small BNN posteriors, but it is prohibitively expensive for networks of size found in modern deep learning applications. For example, Izmailov et al. (2021b) deploy hundreds of Tensor Processor Units to study the mixing and performance of HMC on real-world nets, and find that they mix well in predictive space, although mixing in weight space is worse. For smaller networks, HMC is widely considered the *gold standard* for representing the Bayes posterior.

### B.2. Variational Inference

Variational inference (VI) (Hinton & Van Camp, 1993; Graves, 2011; Blundell et al., 2015; Blei et al., 2017) considers variational distributions from the family $\mathcal{Q} = \{q(\theta; \psi) \mid \psi \in \Psi\}$ to approximate the posterior $\pi(\theta \mid \mathcal{D})$. The metric used is often the "reverse" KL divergence

$$D_{\mathrm{KL}}(q(\theta; \psi) \parallel \pi(\theta \mid \mathcal{D})) = \mathbb{E}_{\theta \sim q(\cdot; \psi)} \log \frac{q(\theta; \psi)}{\pi(\theta \mid \mathcal{D})}. \tag{9}$$

VI casts the (approximate) inference problem as *optimization*, finding the variational distribution minimizing the divergence from the posterior. The objective is equivalent to the objective maximizing the *evidence lower bound (ELBO)*

$$\psi^* \in \arg\min_{\psi} D_{\mathrm{KL}}(q(\theta; \psi) \parallel \pi(\theta \mid \mathcal{D})) \iff \psi^* \in \arg\max_{\psi} \mathcal{L}_{\mathcal{Q}}(\psi; \mathcal{D}), \tag{10}$$

where

$$\mathcal{L}_{\mathcal{Q}}(\psi; \mathcal{D}) = \mathbb{E}_{\theta \sim q(\cdot; \psi)} \log \frac{\pi(\theta) \prod_{i=1}^{n} p(y_i \mid x_i, \theta)}{q(\theta; \psi)}, \tag{11}$$

which no longer depends on the normalized posterior density $\pi(\theta \mid \mathcal{D})$.

Note that the ELBO can be rearranged as

$$\mathcal{L}_{\mathcal{Q}}(\psi; \mathcal{D}) = \sum_{i=1}^{n} \mathbb{E}_{\theta \sim q(\cdot; \psi)} \log p(y_i \mid x_i, \theta) - D_{\mathrm{KL}}(q(\theta; \psi) \parallel \pi(\theta)), \tag{12}$$

providing insight into the optimization objective: the first log-likelihood term encourages the variational distribution to assign high likelihood to the observations, and the second term acts as a regularizer, drawing the variational distribution towards the prior.

For BNNs we need to resort to stochastic gradient-based optimization (Hoffman et al., 2014; Wingate & Weber, 2013). Assuming our variational distribution can be *reparameterized*, i.e., $\theta \sim q(\cdot; \psi)$ can be drawn as $\epsilon \sim \mathcal{N}(0, \mathbf{I}) \sim \phi(\cdot), \theta = T_{\psi}(\epsilon)$ for some $T_{\psi}$ differentiable transform with non-singular Jacobian, parameterized by variational parameters $\psi$. Then the variational density can be computed via change of variable as $q(\theta; \psi) = \phi(\epsilon) \left| \det \frac{dT_{\psi}(\epsilon)}{d\epsilon} \right|^{-1}$. The stochastic ELBO gradient estimate becomes

$$\widehat{\nabla_{\psi} \mathcal{L}_{\mathcal{Q}}}(\psi; \mathcal{D}) = \frac{1}{N} \sum_{j=1}^{N} \nabla_{\psi} \log \frac{\pi(T_{\psi}(\epsilon_j)) \prod_{i=1}^{n} p(y_i \mid x_i, T_{\psi}(\epsilon_j))}{\phi(\epsilon_j) \left| \det \frac{dT_{\psi}(\epsilon_j)}{d\epsilon} \right|^{-1}} \tag{13}$$

where $\epsilon_j \overset{\text{i.i.d.}}{\sim} \mathcal{N}(0, \mathbf{I})$. This can conveniently be computed using automatic differentiation engines.

For BNNs the variational family $\mathcal{Q}$ needs to be simple enough to be able to store the variational parameters $\psi$ in memory, as even quadratic memory requirement in the number of model parameters is prohibitive. The *mean-field* assumption is commonly employed (Blei et al., 2017; Farquhar et al., 2020b) where the variational distribution factorizes $q(\theta; \psi) = \prod_i q(\theta_i; \psi_i)$[2]. In particular, for BNNs the most commonly used variational family is the mean-field Gaussian $q(\theta_i; \psi_i) = \mathcal{N}(\theta_i; \mu_i, \sigma_i^2)$ where $\psi$ consists of a vector of means $\mu^{(q)}$ and a vector of scales $\sigma^{(q)}$.

---

[2]Even though this seems like a restrictive family, enforcing independence, it has been shown to be expressive in deep networks (Farquhar et al., 2020a)

In many cases, the KL term of Equation (12) is available in closed form, allowing one to reduce the variance of the gradient estimator:

$$\widehat{\nabla_\psi \mathcal{L}_\mathcal{Q}}(\psi; \mathcal{D}) = \frac{1}{N} \sum_{j=1}^{N} \sum_{i=1}^{n} \nabla_\psi \log p(y_i \mid x_i, T_\psi(\epsilon_j)) - \nabla_\psi D_{\mathrm{KL}}(q(\theta; \psi) \parallel \pi(\theta)). \tag{14}$$

For example, in the mean-field Gaussian case with mean-field Gaussian prior $\pi(\theta) = \mathcal{N}(\mu^{(\pi)}, \sigma^{2,(\pi)})$,

$$D_{\mathrm{KL}}(q(\theta; \mu_{(q)}, \sigma_{(q)}) \parallel \pi(\theta)) = \sum_i \left( \log \frac{\sigma_{(\pi),i}}{\sigma_{(q),i}} + \frac{\sigma_{(q),i}^2 + (\mu_{(q),i} - \mu_{(\pi),i})^2}{2\sigma_{(\pi),i}^2} - \frac{1}{2} \right). \tag{15}$$

In practice, one needs many ad-hoc tricks and tweaks to make VI work on larger models (Farquhar et al., 2020a). Without these, optimization fails due to the high variance of the stochastic gradients. To reduce this, practitioners have initialized the variational variances to very small values to aid the first stage of optimization, used early stopping criteria, or used the $\beta$-ELBO[3] with a $\beta < 1$:

$$\mathcal{L}_\mathcal{Q}^{(\beta)}(\psi; \mathcal{D}) = \sum_{i=1}^{n} \mathbb{E}_{\theta \sim q(\cdot; \psi)} \log p(y_i \mid x_i, \theta) - \beta D_{\mathrm{KL}}(q(\theta; \psi) \parallel \pi(\theta)), \tag{16}$$

over-weighting the observation likelihood compared to the regularization term, allowing a tighter fit to the data at the cost of less regularization towards the prior.

### B.3. Laplace Approximation

The Laplace approximation (Laplace, 1774) provides a local Gaussian approximation of a distribution around its MAP. It has been widely used as an approximate Bayesian inference algorithm (MacKay, 1992a) both classically and recently in the context of Bayesian deep learning (Daxberger et al., 2021; Schnaus et al., 2023).

Laplace approximation makes a second-order approximation to the log-density of the distribution: given a probability density $f(\theta)$ and a MAP solution $\theta_{\mathsf{MAP}}$,

$$\log f(\theta) \approx \log f(\theta_{\mathsf{MAP}}) + \nabla \log f(\theta_{\mathsf{MAP}})^\top (\theta - \theta_{\mathsf{MAP}}) + \frac{1}{2}(\theta - \theta_{\mathsf{MAP}})^\top \nabla^2 \log f(\theta_{\mathsf{MAP}})(\theta - \theta_{\mathsf{MAP}}) \tag{17}$$

$$= \log f(\theta_{\mathsf{MAP}}) + \frac{1}{2}(\theta - \theta_{\mathsf{MAP}})^\top \nabla^2 \log f(\theta_{\mathsf{MAP}})(\theta - \theta_{\mathsf{MAP}}) \tag{18}$$

assuming the gradient vanishes at the MAP, resulting in the approximation

$$f(\cdot) \approx \mathcal{N}(\cdot; \theta_{\mathsf{MAP}}, [-\nabla^2 \log f(\theta_{\mathsf{MAP}})]^{-1}) \tag{19}$$

once the density is normalized.

While Laplace and Gaussian VI both produce approximate Gaussian posteriors, it is important to note that VI is a *global* approximation whereas Laplace is a *local* approximation. We will see in our experiments how this distinction defines the behaviors of their approximate posteriors.

In the context of Bayesian inference, the likelihood factorizes $\log \pi(\theta \mid y) = \log \pi(\theta) + \log p(y \mid x, \theta) + \text{const}$. Alternatively, to the Hessian of the negative log-likelihood, it is common to use the Fisher information matrix

$$I(\theta) = -\mathbb{E}\left[ \nabla^2 \log p(Y \mid x, \theta) \mid \theta \right] = \mathbb{E}\left[ (\nabla \log p(Y \mid x, \theta))(\nabla \log p(Y \mid x, \theta))^\top \mid \theta \right] \tag{20}$$

or the empirical Fisher

$$\widehat{I}(\theta) = (\nabla \log p(y \mid x, \theta))(\nabla \log p(y \mid x, \theta))^\top \tag{21}$$

at $\theta_{\mathsf{MAP}}$. A further widely used alternative is the generalized Gauss-Newton (GGN) matrix, which is a positive semi-definite approximation to the Hessian, coinciding with $I$ for common observation models (Daxberger et al., 2021; Murphy, 2012).

---

[3]Analogously to the $\beta$-VAE (Higgins et al., 2016).

The size of the Hessian is quadratically large in the number of model parameters, which is prohibitive for real-world BNNs of thousands or millions of parameters. This necessitates further approximations. The simplest is a diagonal approximation to the Hessian or the Fisher information matrix (MacKay, 1992b). Even though it is a restrictive approximation enforcing independence in the posterior, recent work by Farquhar et al. (2020b) found it to remain expressive for deep networks.

Other approximations produce low-rank or block-diagonal Hessian or Fisher matrices. In particular, the popular Kronecker-factored approximate curvature (K-FAC) (Martens & Grosse, 2015; Daxberger et al., 2021; Schnaus et al., 2023) block-diagonal approach further factorizes the per-layer Fishers, which are themselves infeasible to store for large networks.

We note that the diagonal Laplace approximation we evaluate is not the most expressive. However, having run full-rank Laplace experiments on smaller networks where it is feasible, we have seen no reason to assume these qualitative behaviors change for more expressive Laplace approximations, including K-FAC.

### B.4. SWAG

Stochastic weight averaging (SWA)-Gaussian (SWAG) (Maddox et al., 2019; Wilson & Izmailov, 2020) is a heuristic *post-hoc* method of obtaining approximate posteriors of trained deep learning models. Interpreting the loss as the negative log-likelihood, the trained solution is $\theta_{\text{MAP}}$. SWAG performs further gradient descent updates on $\theta$ with a high learning rate $\eta$, exploring the region of the weight space near $\theta_{\text{MAP}}$. We record the trajectory of $\theta$ and fit a low-rank Gaussian to these samples to get the approximate posterior.

Its motivations are arguably Bayesian. These include the local quadratic nature of the loss function around $\theta_{\text{MAP}}$, the same fact that is exploited by Laplace. The $\eta$ hyperparameter controls the posterior variance: the larger the learning rate is, the higher-variance the samples from the trajectory. It has been used by Shwartz-Ziv et al. (2022) in the context of Bayesian deep learning to pretrain a prior for a transfer learning task.

### B.5. SGLD

Stochastic Gradient Langevin Dynamics (Welling & Teh, 2011) is a popular practical Markov Chain-based sampling scheme for BNNs. The continuous-time Langevin diffusion with stationary measure $\pi(\theta \,|\, \mathcal{D})$ is

$$d\theta_t = -\nabla \log \pi(\theta \,|\, \mathcal{D})dt + \sqrt{2}B_t$$

where $B_t$ is a Brownian motion. The score $\log \pi(\theta \,|\, \mathcal{D})$ makes it tractable to deal with the posterior because the normalizing constant cancels out. To simulate this continuous-time dynamics, one traditionally uses Euler discretization schemes

$$\theta_{t+1} = \theta_t - \epsilon_t \nabla \log \pi(\theta_t \,|\, \mathcal{D}) + \sqrt{2\epsilon_t}\xi_t,$$

where $\epsilon_t$ is the learning schedule and $\xi_t \overset{\text{i.i.d.}}{\sim} \mathcal{N}(0, 1)$. For further computational efficiency, one would take stochastic estimates of the gradient by estimating it in batches of size $b$:

$$\theta_{t+1} = \theta_t - \epsilon_t \left( \nabla \log \pi(\theta_t) + \frac{|\mathcal{D}|}{|\mathcal{B}_t|} \sum_{(x,y) \in \mathcal{B}_t} \nabla \log p(y \,|\, \theta; x) \right) + \sqrt{2\epsilon_t}\xi_t.$$

For large batch sizes and small $\epsilon$, we expect the chain to be close to the stationary distribution; however, the discretization and the stochastic gradient estimate both introduce some bias. It is possible to add a Metropolis-Hastings-style accept-reject step to the algorithm to unbias it, but most practical algorithms choose to skip this step for computational efficiency, arguing that the bias is compensated for by decreased variance.

### B.6. Further Bayesian and Non-Bayesian Alternatives

Other Bayesian approximate inference algorithms include Monte Carlo dropout (Gal & Ghahramani, 2016), and further variants of stochastic-gradient Monte Carlo (Chen et al., 2014; Zhang et al., 2020; Wenzel et al., 2020).

There are various further methods for averaging over different models or producing uncertainty estimates that are not traditionally considered Bayesian in their motivation. Deep ensembles (Lakshminarayanan et al., 2017) are state of the art for accuracy and out-of-sample detection on many tasks. They are an ensemble of $K$ different MAP solutions $\theta_{\text{MAP}}^{(k)}$, often

trained with an added repulsive force between them to ensure they converge to different MAP solutions in terms of their generalization behavior. Even though they are not widely considered a Bayesian approach, there have been arguments for considering these discrete model averages as an approximation to the BMA (Wilson & Izmailov, 2020; D'Angelo & Fortuin, 2021)

$$\frac{1}{K}\sum_{k=1}^{K}p(y\,|\,x,\theta_{\mathsf{MAP}}^{(k)}) \approx \int_{\Theta}p(y\,|\,x,\theta)\pi(d\theta\,|\,\mathcal{D}). \tag{22}$$

For modern large-scale models, training multiple MAPs is not feasible, offering a future research direction of more scalable ensemble-based methods.

Other non-Bayesian approaches include epistemic neural networks (Osband et al., 2023) that incorporate a small amount of randomness as input to the network, which allows marginalization of joint predictions.

## C. Martingale Posteriors: A Review

In this section we briefly discuss the martingale posterior framework of Fong et al. (2023), and provide more context on how we apply its ideas to test for Bayesian behavior.

The traditional Bayesian approach is to conduct a prior-posterior calculation on some parameter of interest. Specifically, in the i.i.d. setting, with an observed sample $Y_{1:n}$, we specify a prior density $\pi(\theta)$ and a sampling density $f_\theta(y)$ to derive the posterior $\pi(\theta\,|\,y_{1:n})$. This posterior is then used to compute the predictive density

$$p(y\,|\,y_{1:n}) = \int f_\theta(y)\pi(\theta\,|\,y_{1:n})d\theta$$

The fundamental premise of Fong et al. (2023), however, is that statistical uncertainty arises due to missing observations. If a complete population could be observed, any identifiable parameter of interest would be known precisely. Under this viewpoint, the foundation of Bayesian inference is not to update a prior distribution, but to assign a predictive distribution on the missing observations *conditional on what has been observed*. Uncertainty is then indexed over different possible imputations of the complete dataset.

More specifically, Fong et al. (2023) partition inference into two tasks: simulating the missing future data $Y_{n+1:\infty}$ (or, in practice, $Y_{n+1:N}$ for some sufficiently large $N$), and then recovering the parameter of interest from this complete data. Given the observed $y_{1:n}$, the key component for this pipeline is a sequence of one-step-ahead predictive distributions $\{p(\cdot\,|\,y_{1:i})\}_{i\geq n}$, which simulate an infinite future dataset $Y_{n+1:\infty}$ through a sequential process of *predictive resampling*:

$$Y_{n+1} \sim p(\cdot\,|\,y_{1:n}), Y_{n+2} \sim p(\cdot\,|\,y_{1:n+1}), \ldots, Y_N \sim p(\cdot\,|\,y_{1:N-1}) \quad \text{for } N \to \infty.$$

The parameter of interest, denoted $\theta_\infty$, is then computed from the combined observed and imputed data $\{y_{1:n}, Y_{n+1:\infty}\}$. The distribution of this computed $\theta_\infty$ is defined as the *martingale posterior distribution*.

The underlying justification for this approach applies a result from Doob (1949), which demonstrates an equivalence between Bayesian parametric and predictive uncertainty. In the above example, the authors demonstrate that choosing the standard Bayesian posterior predictive distribution would indeed recover the conventional posterior distribution of the parameter.

### C.1. Predictive Evaluations of Bayesian Updating

The predictive representation of Bayesian inference presented above suggests the possibility of predictive diagnostics to evaluate whether models are respecting the properties of Bayesian inference. For instance, Falck et al. (2024) use this one-step-ahead characterization to probe whether large language models are Bayesian in their in-context learning abilities, answering negatively.

Mlodozeniec et al. (2024) take a similar approach to probe small-scale Bayesian deep learning models via repeated retraining for sequential inference. They evaluate whether various predictive rules are "implicitly Bayesian" by testing whether new predictions are exchangeable with observed data, finding that Bayesian algorithms may fall short on this test in practice.

In the case of BNNs, the martingale posterior framework mirrors existing trends that decouple Bayesian inference from the standard parametric interpretation, and instead takes a functional perspective focused on behavior in predictive space.

However, one-step-ahead predictives correspond to re-training the Bayesian model with an extra (synthetic) observation repeatedly, which is infeasible for at-scale neural network models.

## C.2. Our Adaptation for BNNs

To test the extent of adherence to Bayesian principles in real-world BNN settings, we therefore adapt this framework into a *single-step predictive resampling* procedure, presented in Section 6.

Broadly speaking, our resampling approach corresponds to jointly generating $Y_{n+1:N} \sim p(\cdot \,|\, y_{1:n})$ as $N \to \infty$ (corresponding to the synthetic labeling procedure) instead of the one-step-ahead decomposition. In the asymptotic case, Doob's result implies that the resampled dataset recovers the true setting of the parameters $\theta$. In the finite case, however, we leverage Proposition 6.1's coherence property to see that we recover the initial setting of the parameters *in distribution*. This finite-case predictive coherence result allows us to probe at-scale BNNs by using test functions (the Bayesian model average) to investigate the closeness of the finitely-resampled martingale posterior distribution to the initial distribution.

## C.3. Conditions for the Martingale Posterior to Be Well Defined

In the context of Section 6, most generally, let us consider samples imputed from some predictive $p'(dy^*)$. Assume that for ($p'$-almost) every $y^*$ the updated belief density exists $\pi_{y^*}(\theta)$. Then, by Fubini's theorem, $MP_{Y^*}\pi(\theta)$ is a well-defined density.

Further, we often care about a quantity of interest, $\mathbb{E}_{\theta \sim MP_{Y^*}(\cdot)}h(\theta)$, for example, the BMA of Section A in the case of our experiment. For these martingale posterior predictions to be well defined, we also require that

$$\iint |h(\theta)|\pi_{y^*}(\theta)p'(dy^*)d\theta < \infty.$$

This is satisfied in our classification experiments because the BMA is bounded.

## D. CIFAR and IMDB Experimental Setup

The setups of the CIFAR and IMDB experiments are entirely based on the work of Izmailov et al. (2021b). We deliberately choose a working codebase and replicate the exact settings from one of the prominent papers in the field where Bayesian algorithms are shown to significantly outperform point predictions. This is to ensure that our experimental findings around shortcomings are not merely an artifact of suboptimal hyper-parameterization. Specifically, we follow the hyper-parameters from Table 4 of Izmailov et al. (2021b). We make our fork of their codebase available at github.com/pitukg/bnn_seq_vi/tree/master/bnn_hmc.

## E. Martingale Posterior Experiment: Further Details and Results

### E.1. Estimation

We described the estimation of the empirical martingale posterior distribution in the concrete setting of the CIFAR-10 experiment in Section 6. This section elaborates further on the methodology to sample synthetic labels, and calculate bootstrapped confidence intervals for reproducibility.

Recall that we wish to do predictive resampling of the set of observations $Y^*$ at $x^*$ given the posterior predictive $\int p(\cdot \mid \theta; x^*)\pi'(d\theta)$. Given the conditional independence of the Bayesian setup, the observation model factorizes.

This gives us the following procedure: use the $\pi'$ posterior to sample a setting of the parameters $\theta$, then noise the predictions $f_\theta(x^*)$ through the observation model $Y^* \sim p(\cdot \mid \theta) = p(\cdot \mid f_\theta(x^*))$. In the classification setting, this "noising" means calculating the logits from the sampled model $\theta$, and drawing independent categorical variables according to those logits for each unlabeled observation. For regression, it similarly means noising the predictions with independent variables according to the observation model.

Given access to $n$ i.i.d. synthetic labelings $\{\hat{y}_i^*\}_{i=1}^n$, we approximate a retrained posterior $\hat{\pi}_i''(\theta) = \pi_{(x,y)\cup(x^*,\hat{y}^*_i)}(\theta)$ for each. Then, we want to ensemble these posteriors to produce the empirical martingale posterior. A group of $k$ samples is drawn from the individually retrained synthetic posteriors each $\{\bar{\theta}_i^j \sim \hat{\pi}_i''(\cdot)\}_{j=1}^k$. Under Bayesian inference, according to Section 6, each $\bar{\theta}_i^j$ is distributed as $\pi'$, although they are not independent within groups. Specifically, any integrable test function $h(\theta)$ is consistently approximated by our empirical martingale posterior as

$$\mathbb{E}h(\theta) \approx \frac{1}{nk}\sum_{i=1}^n\sum_{j=1}^k h(\bar{\theta}_i^j).$$

We could pick any (integrable) test function to check the equivalence of the two distributions by comparing its expectation under both. We pick the Bayesian model averaged predictions from Section A. The model-averaged logits of the individually retrained posteriors are averaged to get our empirical martingale posterior BMA.

For VI, we simply sample $n = 20$ settings of the parameters corresponding to ordinal seeds, and $k = 20$ ensembling samples. For SGLD, we take $n = 18$ equally spaced checkpoints post-warmup, from 1450 to 9950, using all $k = 900$ correlated Markov chain samples to ensemble the posteriors. The seeds used to draw the noising variables for labeling images match the seeds for the $\pi'$ samples (for VI), or the checkpoint indices (for SGLD).

To get confidence intervals on our martingale posterior distribution test metrics, we resample and ensemble these individually-retrained synthetic posterior model averages with replacement, yielding a bootstrap confidence interval around the empirical martingale posterior BMA. Concretely, let $\{B_i \overset{\text{i.i.d.}}{\sim} \mathcal{U}[1,n]\}_{i=1}^n$ be random bootstrap indices resampling the individually retrained synthetic posteriors with replacement. The bootstrapped value of a test statistic $h(\theta)$ is given by

$$h^{(B)}(\theta) = \frac{1}{nk}\sum_{i=1}^n\sum_{j=1}^k h(\bar{\theta}_{B_i}^j),$$

whose quantiles give an asymptotic confidence interval for our empirical martingale posterior mean estimator above, because between groups, the $\bar{\theta}_i^j$ are independent.

## E.2. IMDB Experiment

In addition to the CIFAR-10 result of Section 6, we perform the martingale posterior test on the IMDB text classification dataset.

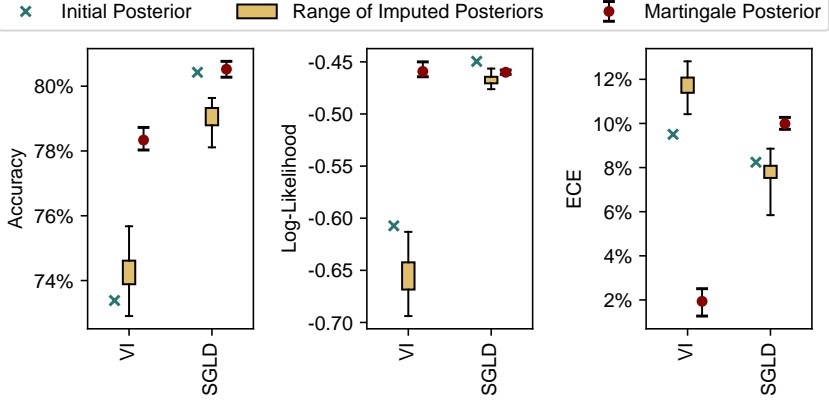

(a) IMDB martingale posterior evaluation.

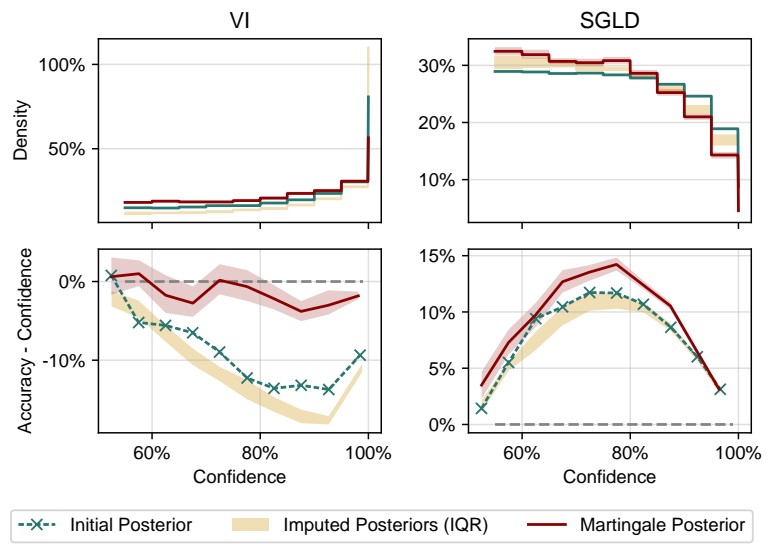

(b) Confidence and calibration of IMDB martingale posterior

Figure 9: **Martingale posterior Bayesian model averages on IMDB splits of size 4000 each.** These plots correspond to Figures 7 and 8, respectively. The VI results are almost identical, strengthening the argument that the martingale posterior can provide performance gains in this setting. For SGLD the performance comparison of the martingale posterior is inconclusive. In this binary classification case SGLD is already very underconfident, and while more ensembling leads to comparable or slightly better accuracy, this does not translate to improved likelihood under this mis-calibrated setting. However, we still observe significant incoherence in uncertainty calibration, which this experiment was originally set up to test.

# F. Sequential Inference Experiment: Additional Results

## F.1. CIFAR-100 Results

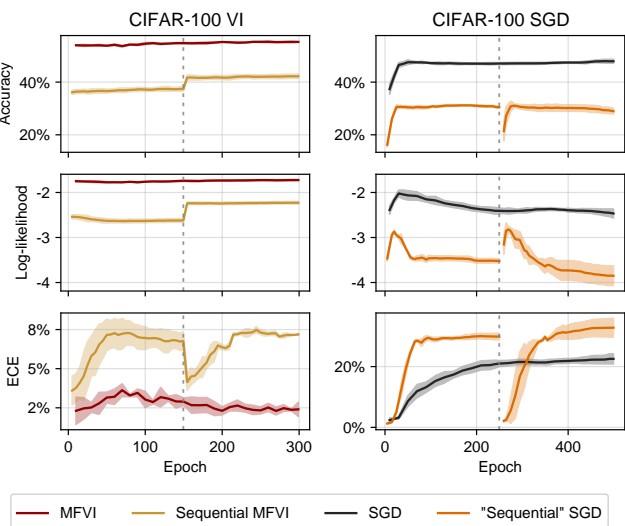

Figure 10: **BMA comparison of single-step and sequentially trained CIFAR-100 MFVI posteriors.** Mean and standard deviation of test metrics from Section A are shown across 3 seeds. The single-step posterior significantly outperforms the two-step iterated posteriors across all three metrics. Unlike CIFAR-10 and IMDB, here the SGD baseline does not show improvement. Although the informative initialization seems to helps at the beginning of training on the second split, the different training hyperparameters may mean this information is lost by the end of its training. Regardless, the performance drop from iterated VI training is even more pronounced than in Figure 4.

## F.2. CIFAR-10 Experiment with PCA Split

We perform the experiment of Section 5.1 on CIFAR-10 a second time, on a non-randomized split along the first PCA direction of the training set. This is analogous to the regression analysis in Section 5.1 where we split the dataset along the $x$ axis. Classical Bayesian learning methods such as Gaussian Processes are able to preserve uncertainty in regions with no observations by falling back to the prior, and we have collected evidence that some Bayesian deep learning methods may not do so. This experiment may suggest potential further inconsistencies on top of the results from Section 5.1, but a conclusive explanation would require more evidence.

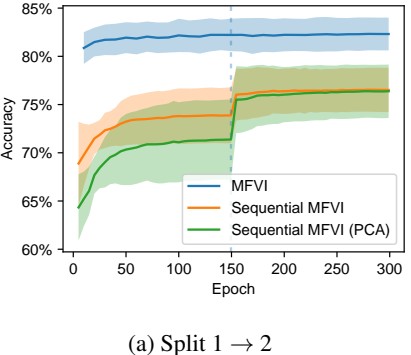
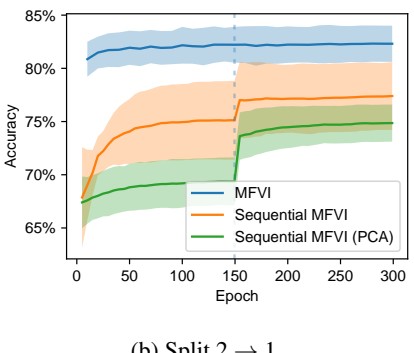

(a) Split $1 \rightarrow 2$                    (b) Split $2 \rightarrow 1$

Figure 11: **Comparing random and PCA-split CIFAR-10 sequential MFVI experiments.** We perform the experiment in two ways: training on the PCA split in the two possible permutations (mean and std across 20 seeds). In the first training step the PCA split underperforms the randomized training, as one might expect. On the second split, however, the results are *inconclusive*: in one of the runs PCA looks worse than random, but in the other run it looks equally as good.

# G. Synthetic Regression Experimental Setup

We make our codebase for the synthetic regression experiments available at `github.com/pitukg/bnn-msc`.

## G.1. Tasks

In our experiments we consider synthetic regression tasks. Even though real-world BNNs are almost exclusively used for classification tasks, there is high relevance of BNN's performance and behavior on regression tasks. The pre-activation of the last layer of classifier architectures (usually a softmax layer) is a low-dimensional (unconstrained) vector that behaves like the output of a regressor network. If BNN regressors fail to respect properties of Bayes, that is an indication of failures for classifiers too, even though they might manifest more gracefully because of the cruder observation model.

The posterior predictive is difficult to visualize if the input is high-dimensional. Therefore, we mainly look at one-dimensional tasks. In the real world, deep learning is used on very high-dimensional tasks, and there is a possibility that BNNs on low and high-dimensional tasks behave differently. We also consider tasks where the one-dimensional underlying input is feature expanded using 100 random Fourier features under a radial basis function kernel (Williams & Rasmussen, 2006). This may provide insight into any differences between BNNs on low-dimensional and medium-dimensional inputs.

We generate toy datasets of the form

$$Y = \sum_{i=0}^{p} \beta_i x^i + \frac{1}{2}\left(\frac{1}{2} + x\right) \cdot \sin(4x) + \varepsilon, \quad \varepsilon \sim \mathcal{N}(0, \sigma^2) \tag{23}$$

for some choice of $p$ and the $\beta_i$, which is complex enough to behave differently in different regions around zero. We observe data in two different groups separated by a gap to assess both interpolation and extrapolation uncertainty of posterior predictives. Because common weight-space priors are not translation nor scale invariant, we consider covariates near zero, and standardize the observations according to standard practice.

## G.2. Architectures

We consider two fully-connected architectures: a small and a larger network. The small network has three hidden layers with 32, 32, and 16 hidden units each, and has 1,714 model parameters. The larger network has 4 hidden layers with 128, 256, 128, and 64 hidden units each, and has 74,690 model parameters, still short of the number of parameters of many modern networks.

The non-linearity used is SiLU ("softplus", "swish", $z \mapsto z/(1 + e^{-z})$), which gives a smoother posterior than ReLU and is standard in Bayesian deep learning.

The observation model, unless otherwise specified, is *heteroskedastic*: the network outputs a mean $\mu_\theta(x)$ and a scale value $\sigma_\theta(x)$ parameterized by network parameters $\theta$, and the output is observed as $Y \mid x, \theta \sim \mathcal{N}(\mu_\theta(x), \sigma_\theta^2(x))$. The last non-linearity of $\sigma_\theta(x)$ is scaled by 0.2 to control the prior observation noise levels compared to the predictive mean, and a small number $10^{-2}$ is added to it, which helped control the variance of the stochastic estimators for ELBO gradients and predictive log-likelihood by avoiding vanishing observation likelihoods.

## G.3. Priors

In line with standard practice (Fortuin, 2022) we use an isotropic Gaussian prior on weights and biases of the network unless otherwise stated. We specify the variances according to the He initialization (He et al., 2015): $w_{ij}^\ell, b_i^\ell \sim \mathcal{N}(0, \frac{2}{D_\ell})$ where $D_\ell$ is the dimension of the activation into layer $\ell$. This ensures that the variance of activations stays constant across the layers.

Note that the isotropic Gaussian prior is not translation invariant in predictive space, in contrast with e.g., Gaussian process priors. Away from zero, their predictives tend to flatten off. Therefore, in line with standard practice, we standardize the input features.

## G.4. Inference Algorithms

To carry out our experiments we use NumPyro (Phan et al., 2019; Bingham et al., 2019), a probabilistic programming library in Python built on JAX (Bradbury et al., 2018). NumPyro readily supports efficient HMC and stochastic variational inference on probabilistic programs.

- **HMC.** For HMC inference, we use the adaptive No-U-Turns (NUTS) sampler (Section B.1), running four chains in parallel, each initialized to independent MAP solutions. For larger networks, this choice of initialization proved crucial, often not mixing when initialized randomly, hinting at the fact that certain parts of the weight space are ill-conditioned, and HMC does not explore the whole space.

  After a burn-in, we use 150 samples from each chain to make predictions, i.e., 600 samples in total.

- **VI.** We use mean-field Gaussian VI, as in Section B.2. In our the KL term of Equation (12) between the prior and variational distribution is analytic, as both are Gaussian, so NumPyro can use the stochastic gradient estimate of Equation (14).

  To deal with the high variance of the gradient estimates, in line with standard practice (Section B.2) we use a number of tweaks:

  - We initialize the variational variances to small values ($10^{-5}$) so that $\mathrm{Var}\big(\nabla_\psi \widehat{\mathcal{L}_{\mathcal{Q}}^{(\beta)}(\psi; \mathcal{D})}\big)$ is small initially. Intuitively, $\mathrm{Var}\big(\nabla_\psi \widehat{\mathcal{L}_{\mathcal{Q}}^{(\beta)}(\psi; \mathcal{D})}\big)$ is small if $\mathrm{Var}\big(\widehat{\mathcal{L}_{\mathcal{Q}}^{(\beta)}(\psi; \mathcal{D})}\big) = \mathrm{Var}_{\theta \sim q(\cdot; \psi)}\big(\sum_{i=1}^{n} \log p(y_i \mid x_i, \theta)\big)$ is, which in turn is small when $\mathrm{Var}_{\theta \sim q(\cdot; \psi)}(\theta)$ is small.
  - We also use momentum through the Adam (Kingma & Ba, 2014) optimizer, and gradient clipping to smooth out the optimization trajectory.
  - Lastly, we use the $\beta$-ELBO (Equation (16)) with a suitable $\beta < 1$ as the maximization objective. This is necessary for more stability during optimization, and better posterior fit (Farquhar et al., 2020a).
    We pick $\beta = 0.325$ for the small network and $\beta = 0.05$ for the larger network, based on a grid search comparing the posterior predictive uncertainty of the resulting posterior to the HMC approximation.

- **Laplace.** We use the diagonal approximation to the empirical Fisher information matrix $\widehat{I}(\theta_{\mathsf{MAP}})$ from Section B.3. Although many in practice use the more expressive K-FAC approximation, there are cases to be made for the diagonal approximation (Farquhar et al., 2020b), and we have not found even the full-rank Hessian or empirical Fisher to give qualitatively different posteriors.

  To increase the stability of the curvature, we need to add a shrinkage factor $\lambda$ to the precisions. The diagonal of the curvature of the loss function $l(\theta)$ is

  $$\left(\left(\frac{\partial l(\theta)}{\partial \theta_1}\right)^2, \ldots, \left(\frac{\partial l(\theta)}{\partial \theta_p}\right)^2\right)^\top.$$

  Adding the shrinkage term and inverting element-wise, we get the scales of the isotropic Gaussian posterior:

  $$\sigma_i = \sqrt{1 \Big/ \left(\frac{\partial l(\theta)}{\partial \theta_i}^2 + \lambda\right)}.$$

  The added shrinkage decreases the isotropic Gaussian parameter scales, making the posterior more confident in the MAP solution.

  It is also related to the cold posterior phenomenon, artificially making the posterior approximation sharper by decreasing the variance of the Gaussian in each direction. We pick $\lambda = 300$ for the small network and $\lambda = 2000$ for the large network, based on a grid search comparing the posterior predictive uncertainty of the resulting posterior to the HMC approximation.

- **SWAG.** Our SWAG implementation relies on the Optax SWAG library (activatedgeek, 2023). 400 samples are collected during high learning-rate gradient descent, and a rank-20 Gaussian is fitted to them. Based on similar grid search tuning, $\eta = 0.02$ for the small network and $\eta = 0.01$ for the larger network.

## H. Synthetic Regression: Tractable Setting

As a sanity check, we compare the approximate posteriors produced by our BNN algorithms to the true analytic posterior in a Bayesian linear regression setting. To keep higher input dimensionality, we expand into 20 random Fourier features with a *squared exponential kernel*, approximating a (sinusoidal) Gaussian process regression.

Recall that Bayesian linear regression with *known observation noise* admits closed-form posterior updates under a Gaussian prior. Let $\boldsymbol{\Phi}(\mathbf{X}) \in \mathbb{R}^{N \times D}$ denote the random Fourier feature matrix for inputs $\mathbf{X}$, with $\phi_j(\mathbf{x}) = \sqrt{2/D} \cos(\mathbf{w}_j^\top \mathbf{x} + b_j)$, where $\mathbf{w}_j \sim \mathcal{N}(0, \kappa^{-1}\mathbf{I})$ and $b_j \sim \text{Uniform}(0, 2\pi)$. We assume the generative model:

$$\text{Prior:} \quad \boldsymbol{\beta} \sim \mathcal{N}\left(\boldsymbol{\mu}_0, \boldsymbol{\Sigma}_0\right),$$
$$\text{Likelihood:} \quad \mathbf{y} \mid \mathbf{X}, \boldsymbol{\beta} \sim \mathcal{N}\left(\boldsymbol{\Phi}(\mathbf{X})\boldsymbol{\beta}, \sigma^2 \mathbf{I}\right),$$

where $\sigma^2$ is fixed. By conjugacy, the posterior distribution over weights $\boldsymbol{\beta}$ is Gaussian:

$$\boldsymbol{\beta} \mid \mathbf{y}, \mathbf{X} \sim \mathcal{N}\left(\boldsymbol{\mu}_n, \boldsymbol{\Sigma}_n\right),$$
$$\boldsymbol{\Sigma}_n = \left(\boldsymbol{\Sigma}_0^{-1} + \frac{1}{\sigma^2}\boldsymbol{\Phi}(\mathbf{X})^\top \boldsymbol{\Phi}(\mathbf{X})\right)^{-1},$$
$$\boldsymbol{\mu}_n = \boldsymbol{\Sigma}_n \left(\boldsymbol{\Sigma}_0^{-1}\boldsymbol{\mu}_0 + \frac{1}{\sigma^2}\boldsymbol{\Phi}(\mathbf{X})^\top \mathbf{y}\right).$$

This analytic solution provides a ground-truth baseline for evaluating approximate inference methods. In our experiments, we set $\boldsymbol{\mu}_0 = \mathbf{0}$ and $\boldsymbol{\Sigma}_0 = \mathbf{I}$.

We conclude that, unlike in the at-scale experiments throughout the paper, in this small classical Bayesian problem, even our approximate inference algorithms such as MFVI adhere to Bayesian properties, even if their approximate posteriors are not faithful (Figure 13). This further underlines that Bayesian-inspired inference in large-scale Bayesian deep learning models leads to different behaviors compared to classical settings, and the benefits of a Bayesian approach do not transfer by default.

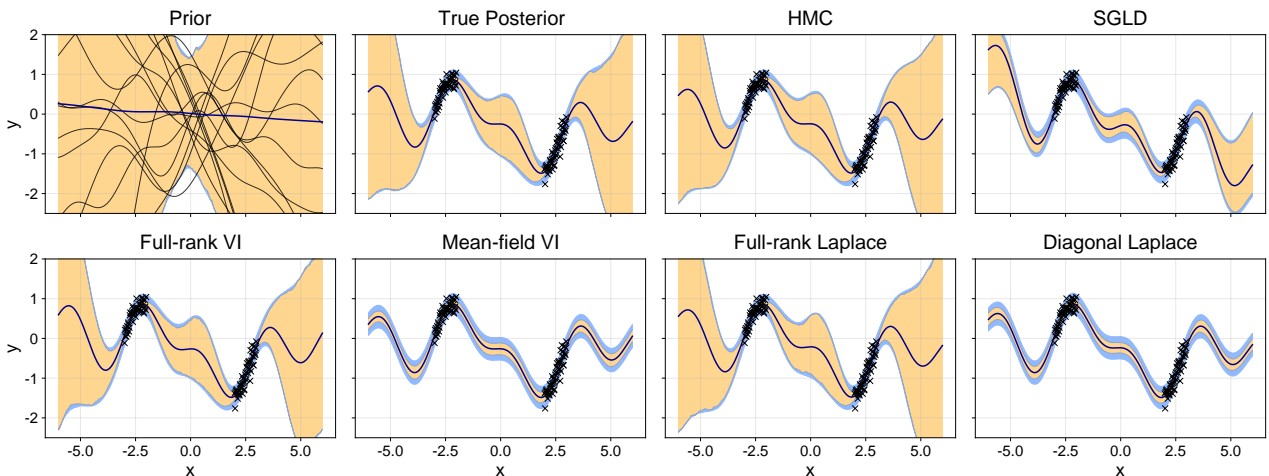

Figure 12: **Comparison of approximate BNN posteriors to the true posterior in Bayesian linear regression.** At this scale HMC seems to match the true posterior. SGLD captures increased uncertainty far away from data. Full-rank Gaussian approximations match the posterior, itself a (full-rank) Gaussian, but diagonal approximations are qualitatively very different.

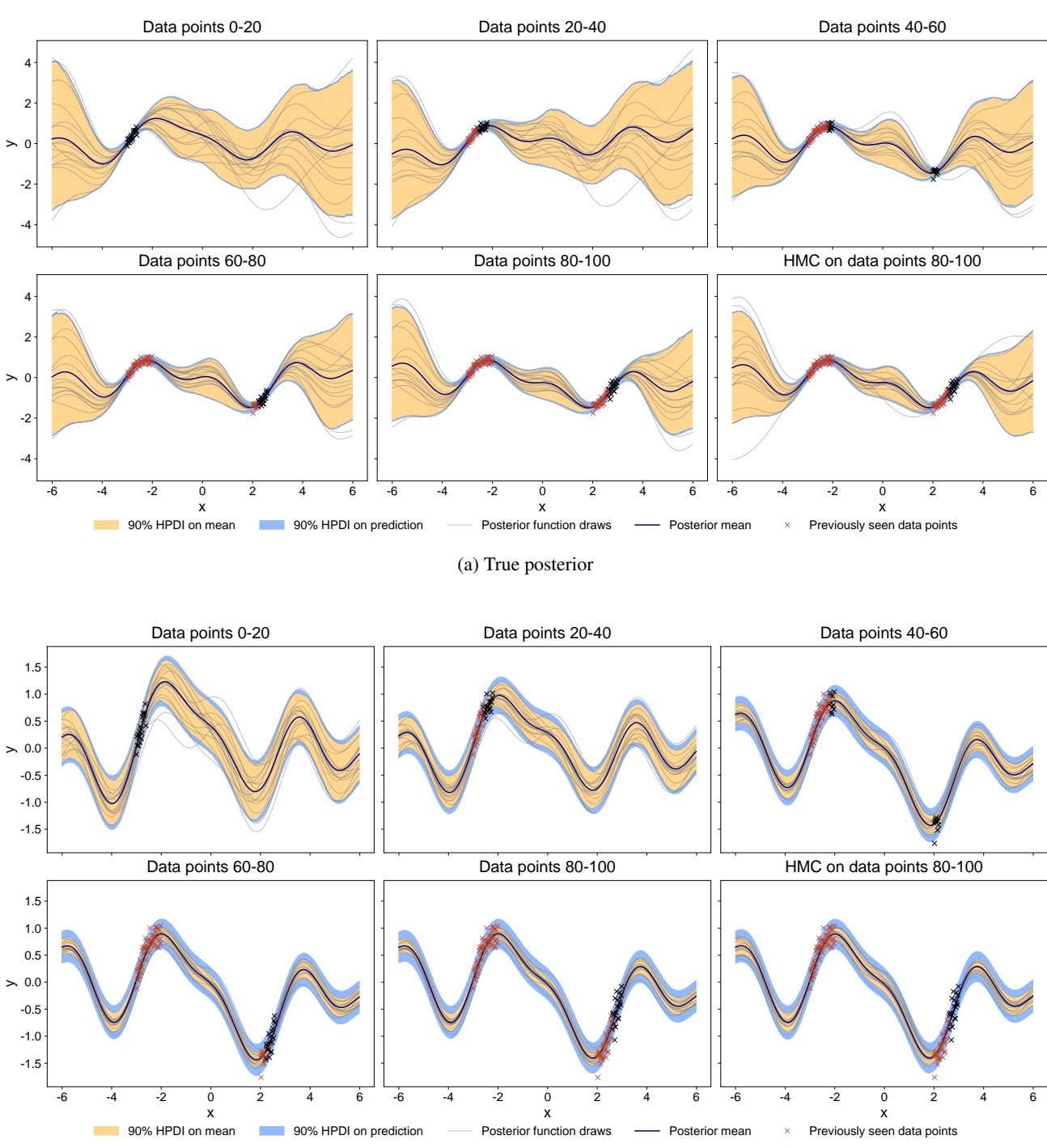

Figure 13: **Comparing MFVI to analytic posterior in sequential inference.** The setting replicates that of Figure 3. Exact inference demonstrates the desired coherent behavior. MFVI updating still shows coherence, despite its apparent lack of faithful posterior representation. For this classical Bayesian setup, even our inexact approximate inference schemes exhibit Bayesian properties. This contrasts the BNN behavior we find at scale.

## I. Synthetic Regression: Homoskedastic Observation Models

For the purposes of the synthetic regression experiments presented elsewhere in the paper, we made the choice to use a heteroskedastic observation model $Y \mid x, \theta \sim \mathcal{N}(\mu_\theta(x), \sigma_\theta^2(x))$ where $f_\theta(x) = (\mu_\theta(x), \sigma_\theta^2(x))$. This choice was made to increase model complexity to get closer to large-scale problems that Bayesian deep learning solutions will encounter in practice.

We recognize that estimating variance is inherently more difficult than estimating the mean, see e.g. Detlefsen et al. (2019). Therefore, in Figure 14 we repeat our sequential experiment from Section 5.1 under *homoskedastic* observation models.

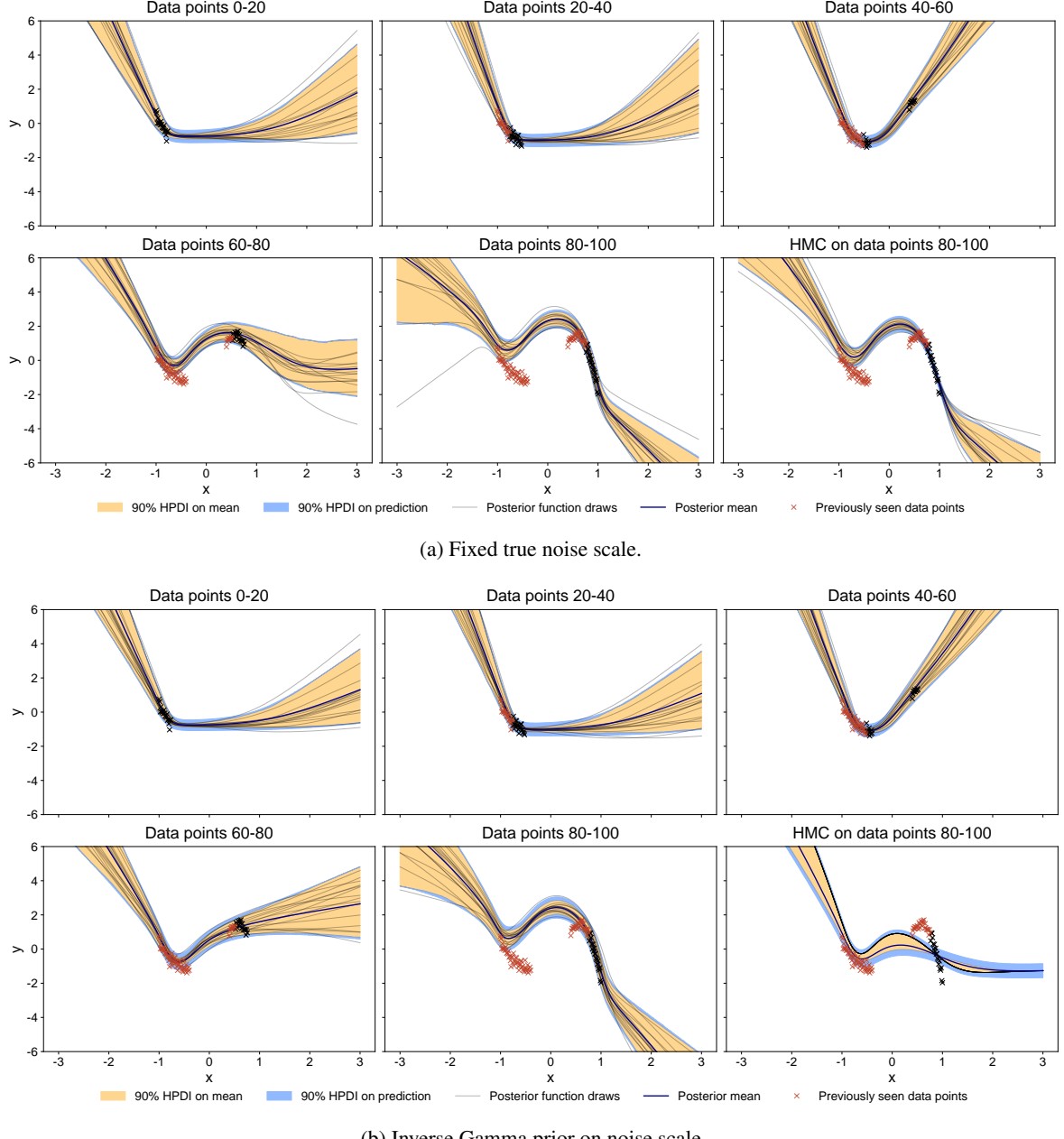

(a) Fixed true noise scale.

(b) Inverse Gamma prior on noise scale.

Figure 14: **MFVI sequential inference under homoskedastic observation models.** In both cases, the results agree with the conclusions of Figure 3.

## J. Synthetic Regression: Further Results

In this section we present further results regarding our synthetic regressions experiment for completeness. In particular, we give evidence for the robustness of our regression results to scaling the network architecture via larger hidden layers or the input dimension random Fourier feature (RFF) expansion.

### J.1. Posterior Uncertainty for Larger Networks

Recall from Section G that we perform our experiments on two network architectures with a smaller and a larger network size, and also on a Fourier feature-expanded version of those architectures. Here we begin presenting these complementary results.

In Figure 15 HMC appears to produce uncertainties in line with our expectation, reducing near the data, and increasing where there have been no observations, including in-between the two groups of observations. VI and Laplace also produce somewhat acceptable uncertainty, but their BALD scores highlight differences in their uncertainty from HMC. SWAG fails to produce sensible uncertainty, and its BMA even fails to fit the data well.

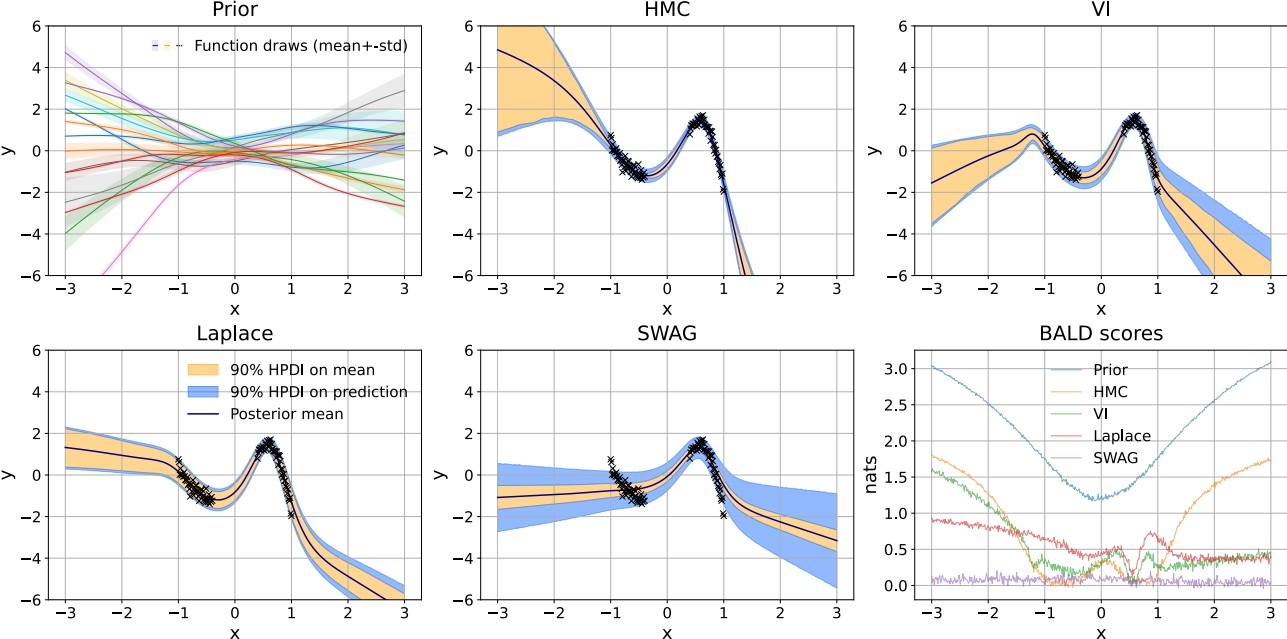

Figure 15: **Larger network approximate posteriors trained on one-dimensional toy regression task.** They are represented by the posterior predictive means, 90% credible sets for the mean representing *model* uncertainty, 90% credible sets for the observation representing full predictive uncertainty, and their BALD scores (measured in natural units of information) giving insight into underlying weight-space uncertainty.

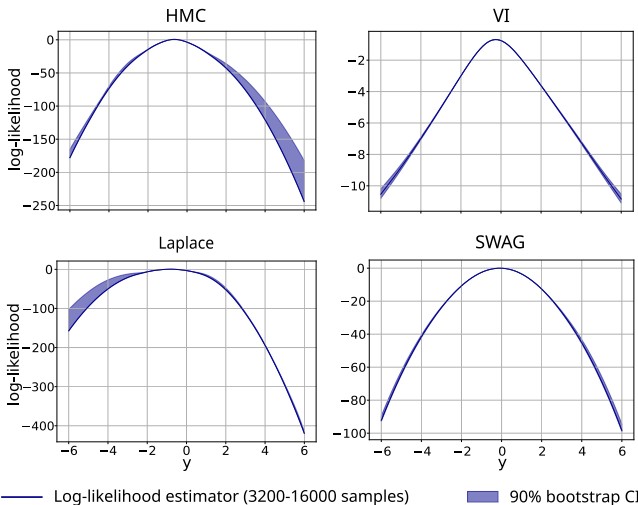

Figure 16: **Cross-sectional predictive log-likelihoods of approximate posteriors from Figure 15.** We use the estimator from Equation (6) to compute predictive log-likelihoods at $x = 0$. The results strengthen the finding of Figure 6: the VI posterior being much heavier tailed than HMC, and Laplace and SWAG posteriors being slightly lighter tailed.

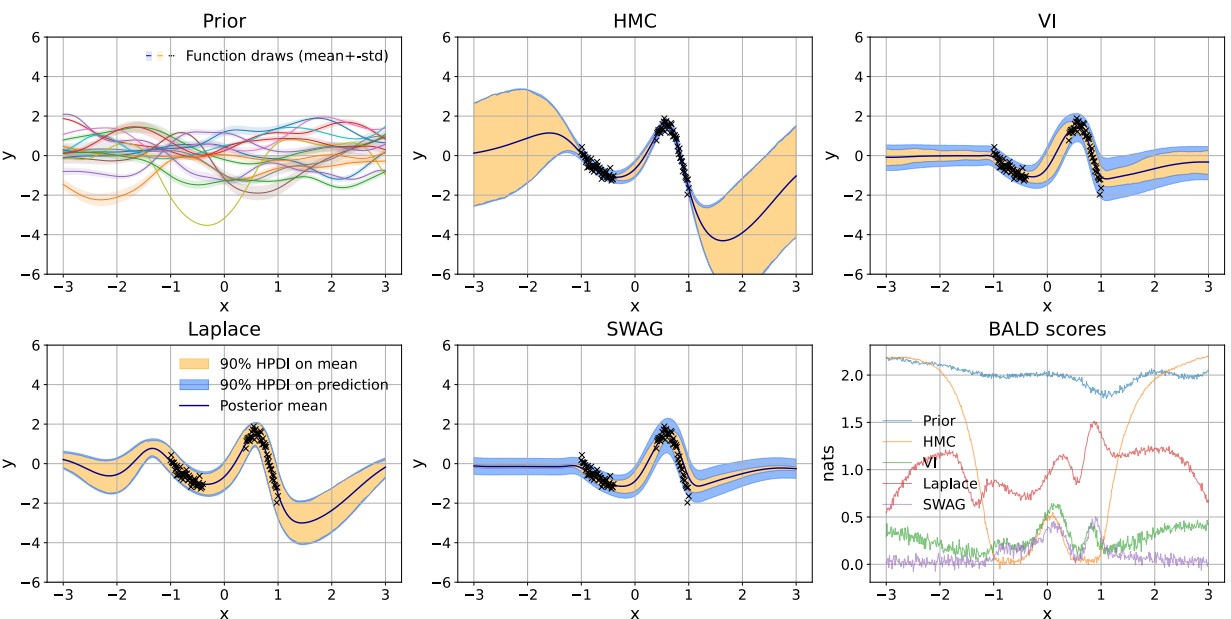

Figure 17: **Feature-expanded smaller network approximate posteriors trained on one-dimensional toy regression task.** The input $x$ is expanded into 100 random Fourier features (RFF) from the RBF kernel, inducing Gaussian Process-like predictives. Again, the HMC approximate posterior uncertainty looks sensible; VI uncertainty is somewhat reasonable but attributes more importance to observing at $x = 1$. The Laplace approximation generalizes very differently from others, and SWAG fails to capture pointwise uncertainty in a sensible way.

## J.2. Sequential Inference

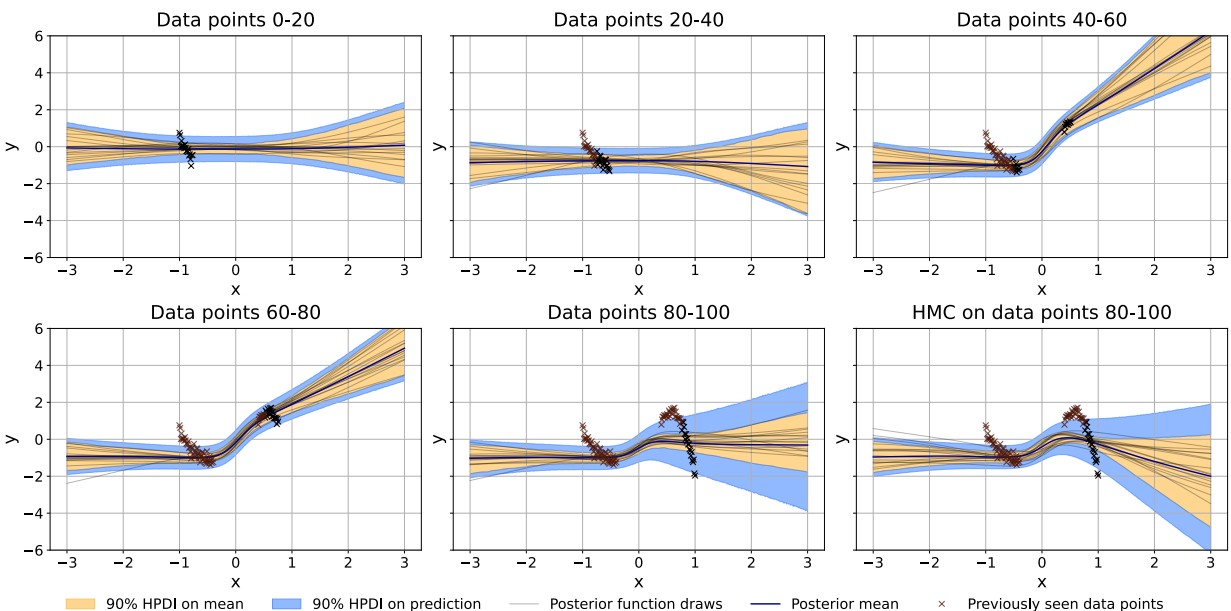

Figure 18: **Smaller network sequential VI inference.** The first step of inference seems to over-weight flat functions and not put enough weight on tighter-fitting functions, causing later inference steps (even HMC) not to fit the first group of observations.

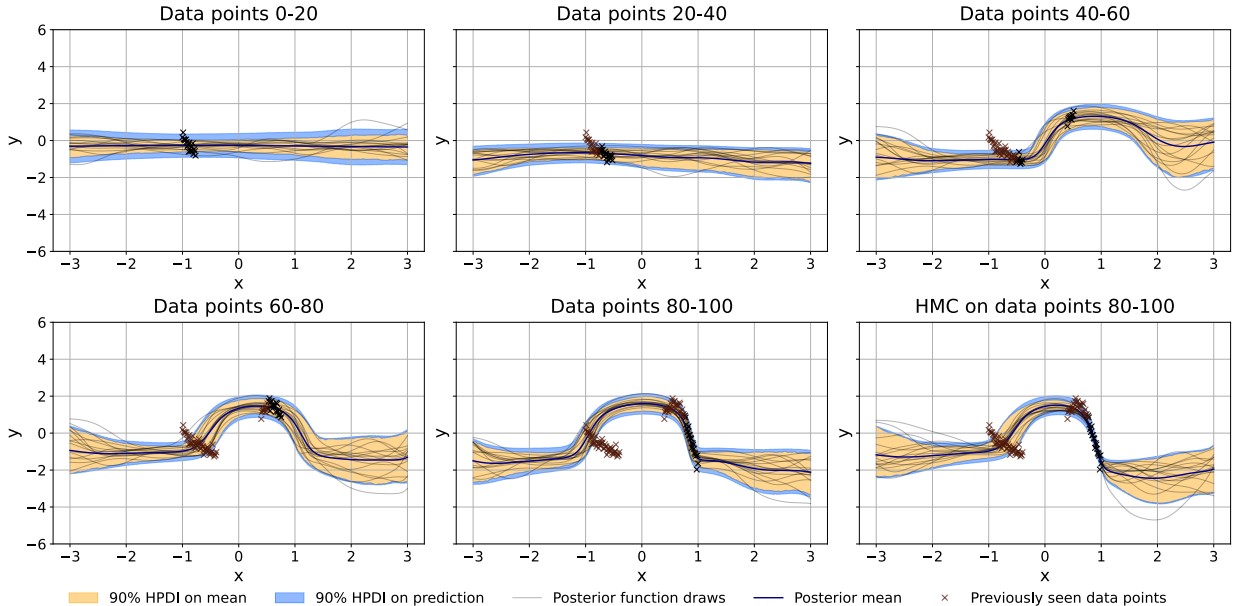

Figure 19: **Larger network sequential VI inference with RFF expansion.** Similarly to Figure 18, the first step of inference seems to introduce too much inaccuracy for later steps to fit the first half of the data particularly well.

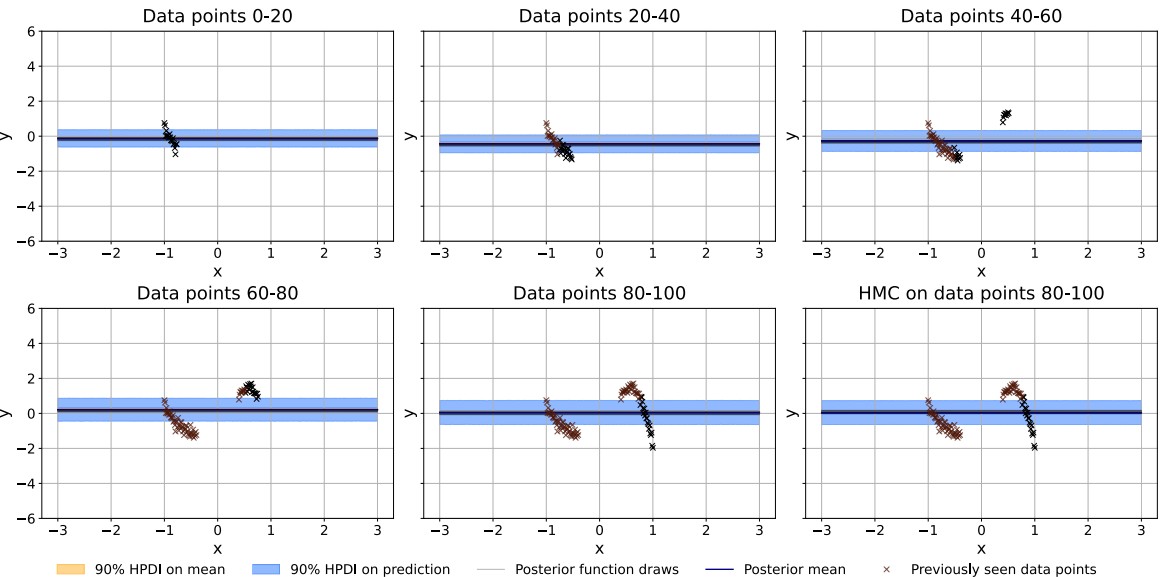

Figure 20: **Small network Laplace sequential inference.** The MAP is flat, and Laplace puts weights on models close to the MAP. The prior after the first step of inference is so restrictive that the subsequent MAPs stay flat too, and even HMC cannot put mass on non-flat functions. This shows complete failure of Laplace to represent the posterior faithfully in this particular case.

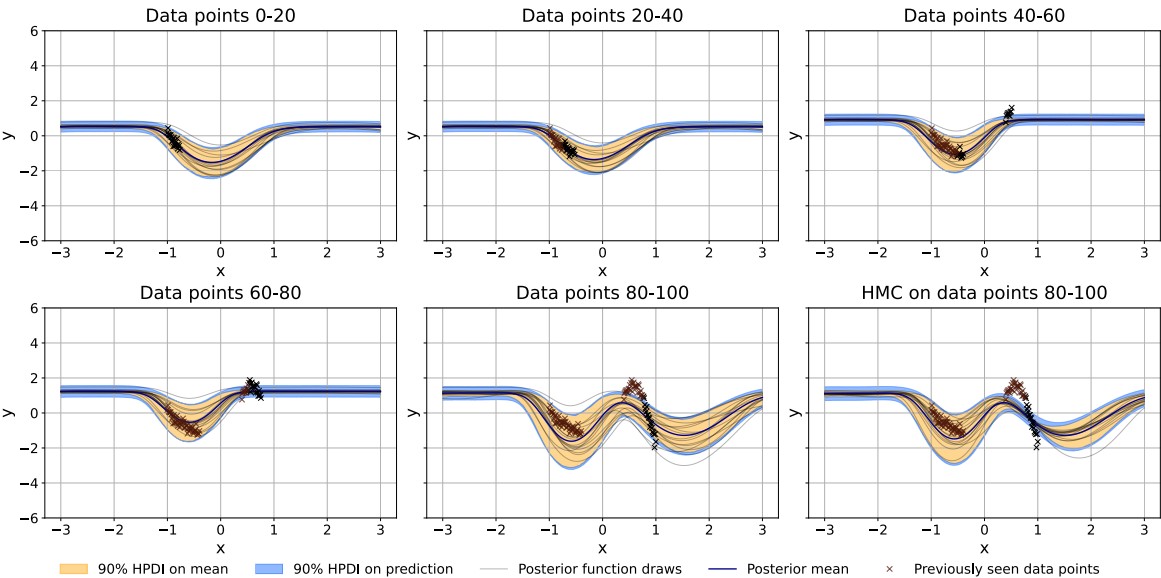

Figure 21: **Small network Laplace sequential inference with RFF expansion.** The feature expansion allows the functions to be richer, and aims to generalize this demonstration to higher-dimensional settings. The small network is chosen so that the MAP is no longer flat under the prior induced by the network. The random Fourier features are taken from the RBF kernel, hence the similarity to kernel ridge with the return-to-zero behavior. Uncertainties do not respect presence of data, and catastrophic forgetting still occurs, even though weaker than without feature expansion.

### J.3. Dataset Admitting a Flat MAP

Here we demonstrate a further pathology of approximate inference in our synthetic regression setup. We pick a range of $x$ values from our regression data that are clustered around zero, where we may reasonably expect our posterior to mix over

two classes of functions, but some of our algorithms fail to do so.

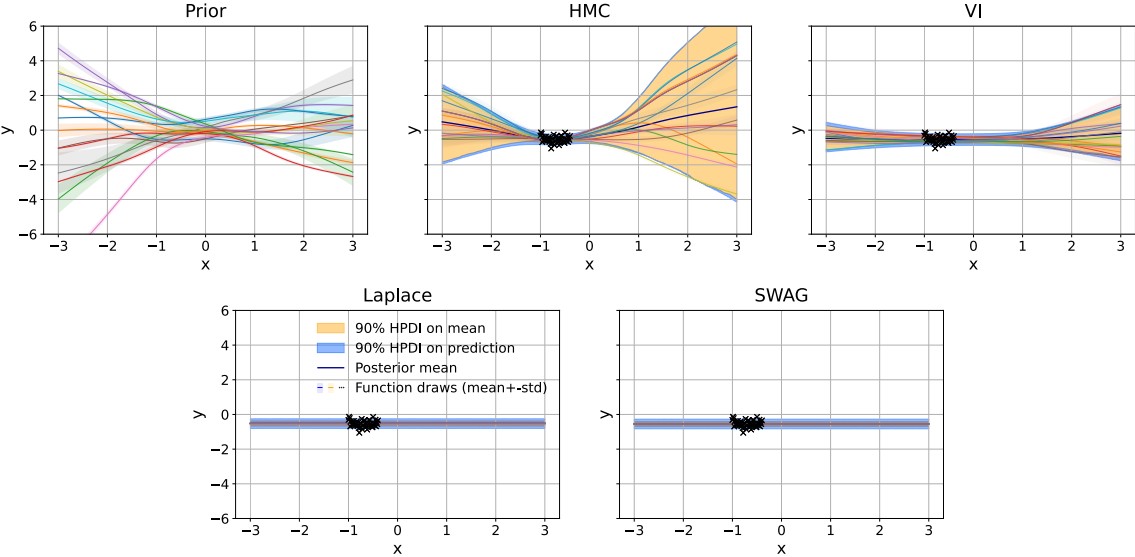

Figure 22: **Posterior function draws on dataset admitting both flat and nonlinear fits consistent with the prior.** One might reasonably expect the posterior to mix over flat functions with higher observation scale and non-linear functions with smaller observation scale. The HMC approximate posterior is mostly in line with this expectation, so it VI although with smaller point uncertainty, however, Laplace and SWAG approximate posteriors fail to represent any non-linear fits, the local approximation potentially breaking down in an ill-conditioned flat region of the parameter space.

### J.4. Intermediate Posterior Forgetting Experiment with VI, and Its Dependence on $\beta$

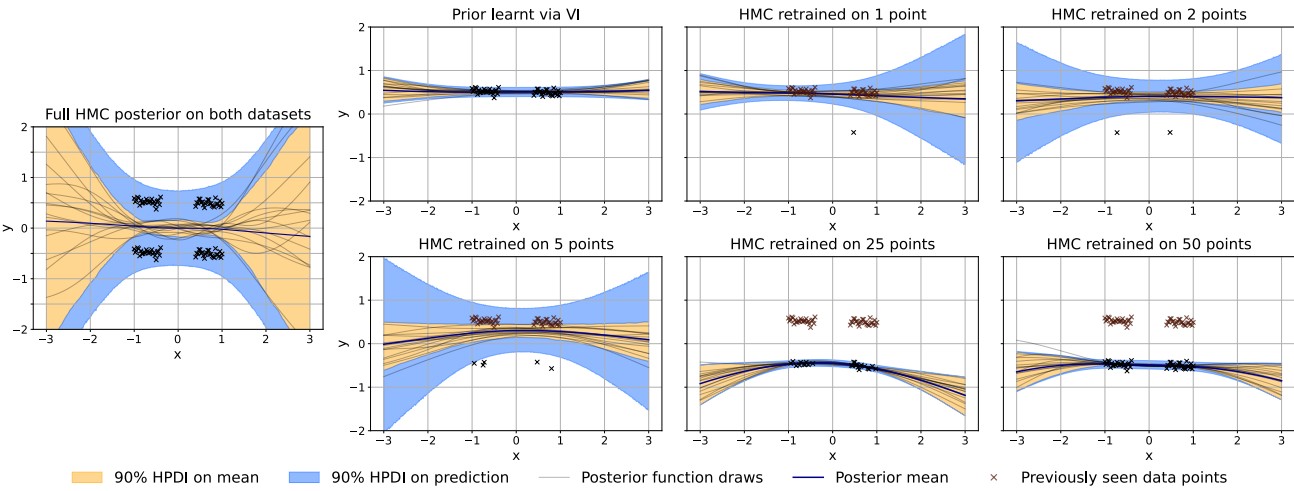

Figure 23: **Pretraining larger network with mean-field VI with tuned $\beta = 0.05$.** The experiment in Section 5.2 is repeated using mean-field VI to pretrain the intermediary posterior. Subsequent HMC inference is performed with this intermediate posterior as prior, observing data from a shifted distribution. The initial VI posterior appears somewhat reasonable here: it puts weights on functions that match the pretraining data, but keeps some amount of uncertainty elsewhere, even though there is no substantial increase in uncertainty in between observations. As we observe new points with the intermediate posterior, the observation noise variance of the likely models increases. However, after only 25 points the original data is fully forgotten, inconsistent with the fact that it should take 50 fresh data points to shift the posterior mean to zero. The single-step HMC approximation is able to produce a sensible posterior mean around zero.

The $\beta$ hyperparameter (see Section B.2, Equation (16)) affects the weighting of information in the VI intermediary posterior: a $\beta < 1$ overweights the likelihood term compared to prior regularization through the scaled KL divergence. In the intermediary posterior, this means a $\beta < 1$ is supposed to over-weight pretraining samples compared to the initial uninformative prior. When HMC inference is performed on this distribution as a prior, one might expect the HMC posterior to favor the original pretraining observations over new data. However, in this case, they are still weighted less compared to fresh observations after HMC inference: forgetting still happens!

An even smaller value of $\beta$ here corresponds to encoding the pretraining observations even *more strongly* into the intermediary posterior; therefore, we expect forgetting to manifest to a lesser extent. While we do observe smaller values of $\beta$ to weight pretraining observations more, the setting of $\beta < 0.001$ where the weighting is correct leads to otherwise qualitatively inaccurate, ill-regularized posterior. We have found the four HMC chains not to mix for such small values of $\beta$, hinting at ill-conditioning in the resulting VI posterior.

We saw a similar result for Laplace in Section 5.2, where despite the shrinkage of the Laplace approximation towards the MAP model (making the approximate distribution more confident on the MAP behavior, albeit without misweighting observations and priors), forgetting also occurred. This surprising observation further demonstrates that practical VI does not respect information as a Bayesian model is supposed to.

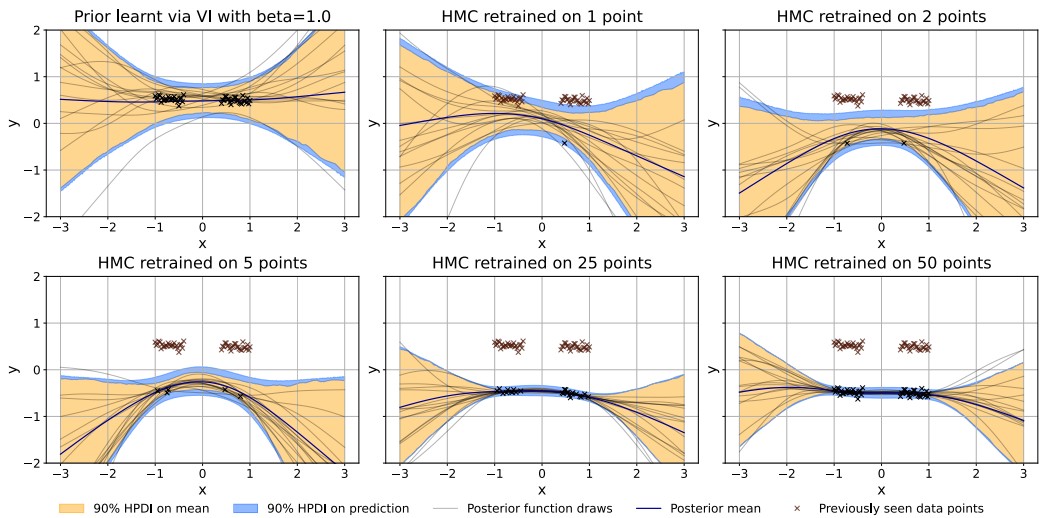

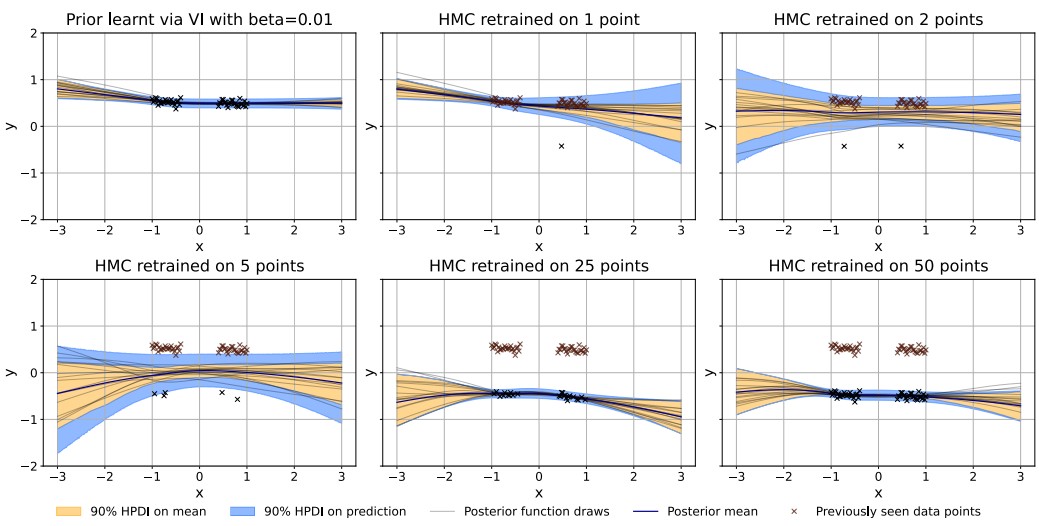

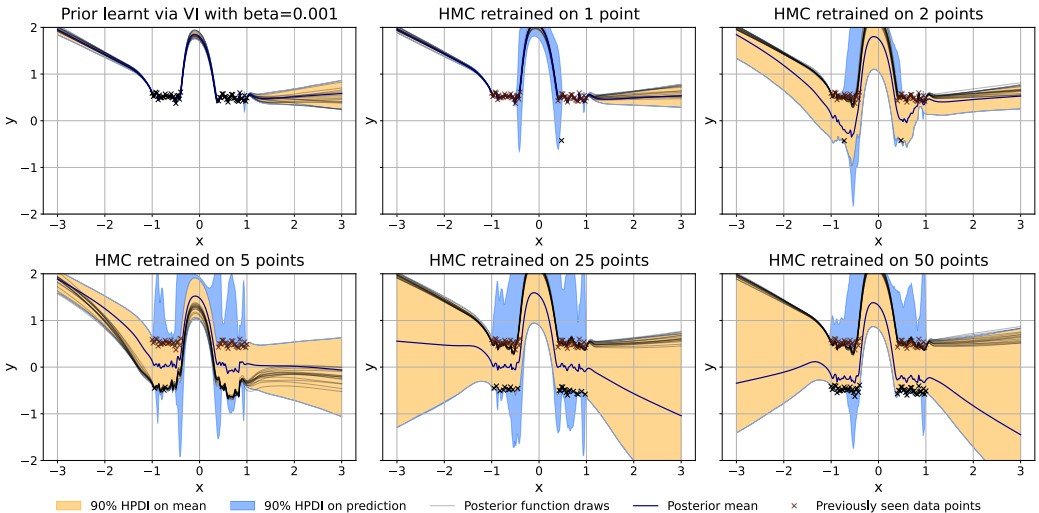

## K. Model Misspecification

To underpin some of the issues in the classical Bayesian literature with suboptimal behavior and pathologies of misspecified Bayesian models, we give a short review and demonstrate empirically that such pathologies can occur for BNNs, too.

Bayes is a principled scheme to update beliefs, which is optimal in the sense defined by Bissiri et al. (2016), provided the model is specified correctly. However, *"all models are wrong"* (Box, 1976), and in general we do not have any guarantees about what Bayes may do under misspecification. In classical statistics there are many theoretical and empirical results about *inconsistency* and the sub-optimality of Bayes on misspecified models (Masegosa, 2020; Grünwald & van Ommen, 2017).

BNNs may also be misspecified. As Bayesian models, BNNs consist of the neural network outputs and the observation model. Even though the underlying neural network is very expressive, it will not be able to *perfectly* represent real-world data-generating processes. For classification, the observation model is given; therefore, we do not expect misspecification to affect classifiers much, but problems may arise, e.g., when making joint predictions. Regression settings are sensitive to the observation model, and in general, we cannot specify the true observation noise distribution, which may lead to inconsistency (Zhang & Nalisnick, 2021).

Section K.2 demonstrates a concrete example where Bayesian inference is inconsistent under misspecification.

### K.1. Robustness to Covariate Shift

*Covariate shift* in a dataset means the distribution of the covariates differs in train and test sets. Even though the observation model conditionally on the inputs might be correctly specified (e.g., classification), it violates the implicit modeling assumption that train and test come from the same distribution. On real-world datasets, covariate shifts occur frequently, as the train set is often carefully curated and augmented, inputs may be noisier at inference time, or usage patterns of an inference system may change over time.

#### K.1.1. IS THE MAP MORE REGULARIZED THAN THE MODEL-AVERAGED PREDICTOR?

The massive empirical success of deep learning suggests that the MAP solution in deep neural nets found by stochastic gradient methods has good generalization performance. This implicit form of regularization has been demonstrated in classical statistics too, e.g., gradient descent on linear regression leads to solutions with small Euclidean norm (Gunasekar et al., 2018).

As we saw in Section 2 the true BNN posterior is a *marginalized* model through BMA. It has been argued in Wilson (2020); Wilson & Izmailov (2020) that this marginalization property is itself the *case* for Bayesian deep learning, making BNNs generalize well in a data-efficient way. But given the well-regularized behavior of the MAP region of the parameter space, the question arises if averaging over models in different regions of the parameter space can lead to loss of robustness compared to the MAP.

Deep ensembles, averaging over multiple MAP solutions, have often outperformed the BMA. In fact, perhaps their success could also be attributed to the favorable generalization behavior of MAPs, when compared to non-MAP models with posterior mass.

### K.1.2. EXISTING EVIDENCE

It has been widely observed, e.g. in Izmailov et al. (2021a;b), that the Bayesian model-averaged predictive is less robust to covariate shift than the MAP. HMC is shown to significantly underperform the MAP solution on datasets like MNIST or CIFAR-10 with corruptions introduced to the test set, even in cases where HMC outperforms MAP on the task without shift.

A concrete example for robustness of MAP is described in Izmailov et al. (2021a). In the simplest case, consider "dead" pixels where an input pixel is always set to zero on the train set. Under most current BNN priors (weights independent with a mode at zero) the MAP solution will set the weights associated with this pixel to zero. However, in the Bayes posterior those weights are distributed according to their prior, therefore the BMA solution depends on the value of this pixel in test. This leads to higher variance in the estimator and may lead to worse generalization.

### K.1.3. SPURIOUS CORRELATIONS EXPERIMENT

To further demonstrate Bayesian model averaging leading to worse robustness, we consider a regression task with *spurious correlations*. We consider a task with standard Gaussian covariates with pairwise correlation $\rho$ in train, but independent in test. The response is generated according to $Y \mid x \sim \mathcal{N}(x_1, \sigma^2)$. We train a BNN on this model.

In this case, all features are informative in train, but the first feature is slightly more informative than the rest. We may expect MAP to put predictive power on the most informative feature, and the Bayes posterior to average over models picking different predictive features too.

Table 1: **Mean squared errors on spurious correlation task.** The use of Laplace prior, equivalent to lasso regularization in linear models, induces sparsity: MAP *selects the optimal model*, picking the first feature only, whereas HMC *averages over the optimal and suboptimal models*. Under Gaussian prior, both solutions pick many features instead of sparsity in the solutions. This also demonstrates the strong, and sometimes counter-intuitive effects that changing to a seemingly similar prior can have on the inference process, highlighting that simple weight-space priors do not capture our prior beliefs well. These results generalize to Bayesian linear regression and other BNN architectures.

|      | LAPLACE PRIOR | GAUSSIAN PRIOR |
|------|---------------|----------------|
| MAP  | **0.0046**    | 0.9059         |
| HMC  | 0.1479        | **0.9045**     |

This experiment demonstrates that the MAP may indeed be more regular, leading to better generalization performance, and converging to the optimal model faster. On the other hand, one might argue that if the BNN prior was to capture our beliefs, the Bayes estimator is decision theoretically optimal under the $L^2$ loss, so its behavior is reasonable. However, the MSE on the shifted dataset is in effect a different loss function. We argue, with today's BNN priors that cannot be said to capture our beliefs well, one should ask which behavior leads to better practical performance in real use cases.

### K.2. Classical Misspecification Experiment: Does Posterior Get Better with More Data?

The Bernstein–von Mises theorem states that the (well-specified) Bayesian posterior asymptotically collapses to the true model. If the true model is, in fact, in the range of candidate models, this consistency property is, of course, desirable.

If the observation model is misspecified, under some conditions the Bernstein–von Mises theorem still holds (Bochkina, 2019), but this collapse leads to *inconsistency*, preventing the inference scheme from converging to a predictive distribution that matches the true distribution the closest (e.g., in KL divergence).

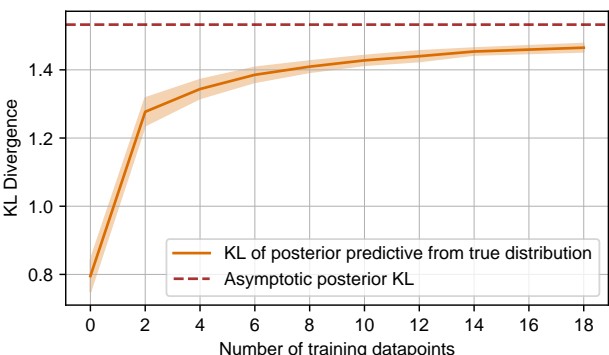

Figure 24: **Divergence of misspecified Bayesian posterior predictive from true generating distribution versus number of observations.** Performance of Bayesian inference *degrades* as more data is observed under a *misspecified* model. As the Bayes posterior collapses to a single model, the posterior predictive gets further and further away from the true data-generating distribution as measured by KL divergence.

To demonstrate the inconsistency of Bayesian inference on a misspecified model, we take the following task. The true data generating distribution is a coin flip between $-1$ and $1$, and we model it as $Y \mid \mu \sim \mathcal{N}(\mu, \sigma^2)$ with $\sigma$ fixed and $\mu \sim \mathcal{N}(0, 1)$.

In this case, the distribution $\mu \sim \mathrm{Bern}(\{-1, 1\})$ minimizes the KL divergence between the true data generating distribution and the posterior predictive distribution. However, simple calculation shows that the Bayes posterior collapses to $\mu = 0$, supported by the simulation in Figure 24 too.

This simple experiment demonstrates that Bayesian inference does not find the optimal distribution over parameters that captures the true distribution the best. In the case of misspecification, we might become overconfident in a badly fitting model. This is worth keeping in mind for BNNs too, especially for regression, where inference is sensitive to the choice of observation model.

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
