# OpenReview forum: "Do Bayesian Neural Networks Actually Behave Like Bayesian Models?"
_ICML.cc/2025/Conference — ICML 2025 poster_

### Official Review · Reviewer_KcGY · 2025-03-10

**Overall Recommendation:** 4

**Summary:**

The paper **"Do Bayesian Neural Networks Actually Behave Like Bayesian Models?"** investigates whether common **approximate inference algorithms** for Bayesian Neural Networks (BNNs) adhere to the theoretical principles of Bayesian belief updating. It empirically evaluates methods such as **Variational Inference (VI), Laplace Approximation, SWAG, and Stochastic Gradient Langevin Dynamics (SGLD)** on synthetic regression and real-world classification tasks.

The key findings indicate that:

### **1. Common BNN Algorithms Lack Functional Variability**
Let $ \theta_s, s=1,\ldots, S $ be a collection of posterior samples from a common BNN algorithm such as Laplace, VI, HMC, etc. We can visualize the function draws $ f_{\theta_s}(x) $ in simple regression settings (e.g., scalar $ x $ and $ y $) by plotting them. Section 4.1 presents evidence that many common BNN methods fail to capture **meaningful uncertainty**, particularly in regions of $ x $ with **few observations**. This suggests that approximate BNN posteriors tend to **over-collapse**, producing function samples that lack diversity and fail to reflect the full range of plausible models. This phenomenon is particularly problematic in **extrapolation regions**, where true Bayesian inference should exhibit greater uncertainty.

### **2. Common BNN Algorithms Forget Past Information in Sequential Learning**
In sequential Bayesian updating, the posterior after seeing two datasets sequentially should match the single-step posterior:

$$
\pi(\theta | D_1) \quad \text{then updating with} \quad \pi(\theta | D_2) \quad \text{should yield} \quad \pi(\theta | D_1 \cup D_2)
$$

Instead, common approximate inference methods exhibit **forgetting** as shown in Section 5.

### **3. Common BNN Algorithms Violate Bayesian Predictive Coherence**
Again let $ \theta_s, s=1,\ldots, S $ be a collection of posterior samples from a common BNN algorithm such as Laplace, VI, HMC, etc. We can use this collection to form a predictive density with which we impute future data. Then on this augmented dataset (original plus imputed) we can reapply the same BNN algorithm to get a new posterior sample. Repeating this imputation $ N $ times, we can get $ N $ new posterior samples. These new samples should remain consistent with the original ones in the sense that metrics such as test accuracy, log-likelihood, and expected calibration error (ECE) should be preserved. However, Section 6 presents evidence that these metrics shift significantly after imputation, which the authors consider to be evidence for predictive incoherence.

## update after rebuttal
I increased my score from 3 to 4 and maintain a positive impression of the paper. The rebuttal was handled well by the authors. I am not convinced by the review that gave a 1 and have decided to increase my score to help the paper's chance of acceptance.

**Claims And Evidence:**

Yes. The main claims—that common BNN algorithms fail to behave in a strictly Bayesian manner—are supported by well-designed empirical analysis.

The claim that martingale posteriors could be a worthwhile pursuit could be further substantiated however.

**Essential References Not Discussed:**

I did not identify any crucial references missing from the discussion. The paper effectively situates its contributions within the existing Bayesian deep learning literature.

**Experimental Designs Or Analyses:**

The experiments span synthetic regression and CIFAR-10 classification to evaluate whether common BNN algorithms exhibit Bayesian properties. They convincingly demonstrate that approximate inference algorithms fail certain Bayesian properties. The empirical design is sound, and additional experiments would likely reinforce rather than alter the conclusions.

However, further experiments  for the martingale posterior could strengthen the paper’s concluding claim that MPs are worthy to pursue in Bayesian deep learning.

**Methods And Evaluation Criteria:**

There are multiple ways in which BNN algorithms may fail to be strictly Bayesian. This paper examines a subset of these properties, focusing on functional variability, forgetting past data, and predictive coherence. The choices are reasonable, as they directly impact certain applications. While the experiments use only synthetic data for regression and CIFAR-10 for classification, this is sufficient to support the claims, as the findings are expected to generalize.

**Other Comments Or Suggestions:**

### Minor comments:
- This paper honestly reads more like a position piece than a purely technical contribution. The only technical contribution is the MP part which is far too brief and preliminary.
- Figure 8 caption: "Their martingale posterior over-reduces confidence"—the phrase "over-reduces" is awkward. Consider rewording for clarity.
- Line 352: "a belief updating scheme $ P(\theta) \mapsto P_y(\theta) $." Since this convention is only used in the context of $ \pi $, why not define it directly in terms of $ \pi $ rather than $ P $?
- The notation for the equation in Line 411 confuses me. On the left-hand side, I don’t understand why there is a hat on $ \pi $.
- Would it be possible to explicitly write out reason for the last equality in the proof of Proposition 6.1?
- In the abstract, the key finding is written as “common BNN algorithms…fail to update in a consistent manner, forget about old data under sequential updates, and violate the predictive coherence properties…” Then later, starting around the last sentence on page 1, it is written that these common BNN algorithms 1) “lack the functional consistency of the Bayesian posterior”, 2) “sequential inference using BNNs fails to propagate information coherently” and 3) “BNN predictive updates are not well-calibrated”. Are these two sets of claims meant to match up one to one and in that order? I fail to see how predictive coherence is the same as well-calibration. Also, I don’t understand the term “functional consistency.” Reading 4.1, it seems that common BNN algorithms lack the true functional “variability”.
- I find the notation used in the CIFAR-10 experiment a bit confusing. Is $ x^* $ a single unlabeled image or a set of unlabeled images?

**Other Strengths And Weaknesses:**

I found this paper well-argued and engaging. It articulates a viewpoint that aligns with my own perspective—that we should not expect BNNs to be strictly Bayesian but instead view them as practical probabilistic models. However, while the paper critiques existing BNN methods effectively, the proposed alternative (martingale posteriors) is presented in a somewhat preliminary manner. The conditions required for the martingale posterior to be well-defined are not fully discussed, and it remains unclear whether it truly resolves the issues identified with standard BNN algorithms.

**Questions For Authors:**

1. The introduction states that BNNs “offer better predictive performance…than deterministic networks in multiple settings.” How do these claims hold up against deep ensembles?

2. The martingale posterior is defined without any conditions on the belief updating scheme. Is this truly appropriate? A belief update could take many forms—does this framework require specific assumptions to guarantee desirable properties?

3. In the CIFAR-10 experiment, why does using the intermediary \( \pi’ \)-predictive to impute labels lead to a well-defined martingale posterior?

4. I’m not sure my reading of the CIFAR-10 experiment is correct. Could you please check my summary of your third claim under Summary and correct any misconceptions I have?

**Relation To Broader Scientific Literature:**

The paper contributes to ongoing discussions in Bayesian deep learning, where much of the focus has been on improving inference algorithms for better predictive performance and uncertainty quantification. While prior works have acknowledged that approximate Bayesian inference does not strictly adhere to Bayesian principles, this paper systematically documents key deviations. The use of martingale posteriors as a potential fix is intriguing but underdeveloped. It would be valuable to compare this to the broader literature on alternative Bayesian approximations and uncertainty estimation techniques.

**Theoretical Claims:**

The paper primarily focuses on empirical analysis, with limited theoretical contributions. Proposition 6.1 formalizes the predictive coherence of Bayesian inference using martingale posteriors. The proof appears correct.

---

> ### Author Rebuttal · Authors · 2025-04-01
>
> Thank you for your careful review and helpful feedback! We are glad you found our paper “engaging” and “well argued”. We hope our responses and new experiments below answer your questions and alleviate the concerns you raised.
> > The proposed alternative (martingale posteriors) is presented in a somewhat preliminary manner ...it remains unclear whether it truly resolves the issues identified with standard BNN algorithms.
>
> While we agree that the paper could be improved by expanding on martingale posterior (MP) evaluations and will present new results on this shortly, we first wanted to discuss its role within our overall contributions. Our primary aim, as you eloquently explain, is to investigate the adherence of BNNs to the theoretical principles of Bayesian updating. Predictive coherence (Section 6) is one of these key principles. The MP framework is a novel tool we propose for this evaluation, which then naturally leads to our experiments in Figures 7 and 8 intended to identify the lack of coherence in existing approaches.
>
> It is only after we have conducted these tests that the avenue of using MPs as a mechanism itself to potentially combat the problem becomes apparent. Given our original aims and the large amount of other content, we thus feel that fully exploring this new idea is not feasible within the scope of the current paper. Instead, we see this as a very exciting avenue for future investigation, adding significance to our work.
>
> Nonetheless, we do plan to significantly expand on our discussion of the MP using the extra page in the camera ready (see also response to Reviewer 8qCC) and an exposition in the appendix for ML audiences, as you very rightly suggest.
> > The claim that [MPs] could be a worthwhile pursuit could be further substantiated …, further experiments for the [MP] could strengthen the paper’s concluding claim that MPs are worthy to pursue in Bayesian deep learning…
>
> To provide more evidence for the generality of the MP findings, __we have repeated our CIFAR experiments on the IMDB dataset__: [see here](https://raw.githubusercontent.com/anon17438/anon-rebuttal-material/main/rebuttal_martingale_posterior_imdb.pdf). The VI results are almost identical, strengthening the argument that the MP can provide performance gains in this setting. For SGLD, while the performance comparison of the MP is inconclusive, we still observe significant incoherence in uncertainty calibration, which this experiment was originally set up to test.
> > It would be valuable to compare [MPs] to the broader literature on alternative Bayesian approximations and uncertainty estimation techniques.
>
> Thank you for raising this. We will add more discussions and references to the broader literature in Section 3 including non-standard “generalized” updates to Bayes’ rule (we assume this is what you mean by “alternative Bayesian approximations"), Gibbs posteriors, PAC-Bayesian methods, and non-Bayesian approaches like epistemic neural networks. Please let us know anything else you think would be important to discuss here.
> ## Questions For Authors
> > 1. The introduction states that BNNs “offer better predictive performance…than deterministic networks in multiple settings.” How do these claims hold up against deep ensembles?
>
> Deep ensembles (DE) have also indeed been shown to offer better performance than deterministic networks, and we actively avoid taking a position on the relative superiority of BNNs and DE. Assessing non-Bayesian uncertainty quantification methods like DE under our Bayesian lens would certainly be an interesting future investigation.
> > 2. The [MP] is defined without any conditions on the belief updating scheme … does this framework require specific assumptions to guarantee desirable properties?
>
> We refer back to our initial discussion that the desirable property of MPs for the purpose of our experiment is to recover the initial posterior. Typically, they will not do so under approximate inference.
> > 3. In the CIFAR-10 experiment, why does using the intermediary $\pi’$-predictive to impute labels lead to a well-defined [MP]?
>
> The regularity assumptions needed for well-defined MPs are generally weak, requiring that the prior and updated belief densities exist, and BMA has finite expectation, clearly satisfied for classification where it is bounded. We will expand on this.
> > 4. I’m not sure my reading of the CIFAR-10 experiment is correct.
>
> The summary is correct. We exploit the property that sampling from $\Theta\sim\pi(\theta|y)$ is equivalent to imputing a random observation $Y'\sim p(y’|y)$ and then drawing the posterior sample $\Theta\sim\pi(\theta|y,y')$. Our experiments find that the samples obtained this way deviate from the original distribution in their BMA, as captured by our evaluation metrics.
>
> We apologize we do not have the space to address all minor comments, but we will incorporate them into the paper.
>
> Again, thank you for the review. Please let us know of any further questions or concerns.

---

### Official Review · Reviewer_yeeB · 2025-03-10

**Overall Recommendation:** 4

**Summary:**

This paper investigates the properties of several popular algorithms for approximate posterior inference in Bayesian neural networks (BNNs). The main experimental findings are that common approximate inference algorithms in BNNs: (a) do not exhibit functional consistency of posteriors, (b) do not propgate information coherently when performing sequential inference, and (c) do not exhibit predictive coherence.

**Claims And Evidence:**

Each of the three experimental findings are supported by evidence.

**Essential References Not Discussed:**

Not to my knowledge.

**Experimental Designs Or Analyses:**

The experimental analyses seem reasonable, although I did not fully understand the martingale posterior experiment.

**Methods And Evaluation Criteria:**

There are a mix of toy and larger-scale experiments, which is sensible to investigate the questions being asked.

**Other Comments Or Suggestions:**

- p. 2, ll. 58-62: Section 5.2 is listed twice.
- p. 3, l. 145 (second column): What does BMA stand for?
- p. 3, l. 151 (second column): It may help the flow to discuss applications of BNNs earlier, in Section 2.
- p. 3, l. 156: Are the sizes listed correctly here? 26 seems like a typo.
- p. 4, l. 190: Has $\beta$ been defined at this point in the paper?

**Other Strengths And Weaknesses:**

I have already described above the main strength, which is to shed light on some problematic issues with approximate posterior inference in BNNs. Soundness and support for claims are also discussed above, and the writing is largely quite coherent.

There are also few weaknesses that deserve mention:
- In a set of experiments like this, it would be nice to have a small version where exact inference is possible. This would help clarify which effects are due to HMC being treated as the gold standard and what is inherent to Bayesian inference itself.
- The paper demonstrates several issues with approximate inference algorithms for BNNs but does not attempt to provide theoretical understanding of why these issues occur or what can be done to remedy them.
- Discussion about key takeaways is also relatively light: should practitioners avoid BNNs completely? Should they be used only in certain contexts? Is more research needed to develop inference schemes that respect e.g. coherence in sequential inference?

**Questions For Authors:**

In addition to the weaknesses raised above: are there any cases where the uncovered pathologies are actually beneficial? One person's incorrect posterior is another person's correct posterior with a different prior. Might there be a case where narrowing down the functions as done by VI, Laplace, and SWAG in Figure 1 can actually help?

**Relation To Broader Scientific Literature:**

The contributions are positioned to explain some folklore knowledge within the Bayesian deep learning community. It is well known that Bayesian methods are sound for models with small numbers of parameters. However, direct application of Bayesian methods to high-dimensional deep neural networks is fraught with practical issues. These issues are known well enough by the community to be explored in the NeurIPS workshop "I Can't Believe It's Not Better (ICBINB)". This paper provides a more systematic study of some of these issues, and offers concrete evidence of the shortcomings of current approximate posterior inference methods.

**Theoretical Claims:**

There are no novel theoretical claims made in the paper.

---

> ### Author Rebuttal · Authors · 2025-04-01
>
> Thank you for your careful review and helpful feedback! We hope that the responses and new results below answer your questions and alleviate any remaining concerns.
> > In a set of experiments like this, it would be nice to have a small version where exact inference is possible. This would help clarify which effects are due to HMC being treated as the gold standard and what is inherent to Bayesian inference itself.
>
> Great suggestion! To provide this, __we have performed an extra experiment with Bayesian linear regression__: [see here](https://raw.githubusercontent.com/anon17438/anon-rebuttal-material/main/rebuttal_exact_inference_experiment.pdf). To keep higher input dimensionality, we expand $x$ into 20 random Fourier features with a squared exponential kernel, approximating a (sinusoidal) Gaussian process regression.
> Exact inference demonstrates the desired coherent behavior. MFVI updating still shows coherence, despite its apparent lack of faithful posterior representation. For completeness, we have run all algorithms in the single-step setting and confirmed that HMC matches the true posterior. For this classical Bayesian setup, even our inexact approximate inference schemes exhibit Bayesian properties. This contrasts the BNN behavior we find at scale. We will expand on this running example to demonstrate the expected behavior with exact Bayesian updating through the paper.
> > The paper demonstrates several issues with approximate inference algorithms for BNNs but does not attempt to provide theoretical understanding of why these issues occur or what can be done to remedy them.
>
> The primary objective of our paper is to highlight the discrepancy between the theoretical expectations and the actual behavior of BNNs. We thus believe it is natural that we focus on experimental evaluation rather than new theory. The theoretical reason for the issues is arguably quite straightforward: inexact inference causes substantial discrepancy in behavior from the true posterior. One could try to do theoretical analysis on this gap itself, but it would likely be challenging and prone to the same theory-practice gap we are investigating in the first place; we feel this is beyond the scope of what can be achieved in the current work.
>
> In terms of remedies, these extremely challenging and fundamental issues at the core of the BNN methodology cannot feasibly be solved in a single paper. We see our work as the first step on this journey, highlighting and carefully documenting the issues, but we expect it to be many years before the community has managed to remedy them, if indeed they ever do.
> > Discussion about key takeaways is also relatively light…
>
> Thank you for this feedback. We will exploit the extra page available for the camera ready to add more discussion on key takeaways. On the specific questions you raise:
> > Should practitioners avoid BNNs completely? Should they only be used in certain settings?
>
> Definitely not! They have been shown to outperform point estimates and provide effective static uncertainty in multiple settings, which our results do not undermine. We simply argue that care is needed in our expectations and interpretation, especially outside these prediction settings where the pathologies we discuss might occur. In particular, we need to be very careful about empirical evaluation, as good static performance may not translate to other desired behaviors.
> > Is more research needed to develop inference schemes that respect e.g. coherence in sequential inference?
>
> We would indeed argue that more research is needed in this direction. Moreover, in our opinion this should not be thought of as simply an inference scheme problem: entirely not Bayesian methods might in practice give more coherence under sequential inference. Even though the problems we raised are a direct result of inexact inference, improving inference schemes is not necessarily the solution: accurate inference may be unachievable, and strategies focusing on final behavior could be more effective. In short, this remains a highly unsolved problem with many possible future research directions.
>
> Other questions we will discuss are _Why does it matter that BNNs are not truly Bayesian?_ and _If BNNs are not Bayesian, what are they?_.  Let us know if there are any other suggestions.
> > p. 3, l. 156: Are the sizes listed correctly here? 26 seems like a typo.
>
> Well caught! Indeed this should say 256.
> > Are there any cases where the uncovered pathologies are actually beneficial?
>
> There is never likely to be any direct benefit in the inconsistent behavior itself, but the deviations from the “correct” posterior may potentially sometimes be beneficial from a static prediction perspective. For instance, [1] found the calibration of VI to be consistently better than more faithful Bayesian approximations such as HMC.
>
> Thank you again for your review and helpful suggestions!
>
> [1] Izmailov et. al. (2021). _What are Bayesian Neural Network posteriors really like?_ ICML

---

> > ### Comment · Reviewer_yeeB · 2025-04-03
> >
> > Thank you for your response and experiments. I do not have any further questions.

---

### Official Review · Reviewer_8qCC · 2025-03-16

**Overall Recommendation:** 3

**Summary:**

The paper investigates the alignment of Bayesian neural networks wrt rigorous Bayesian principles/ideals. To do so, tasks like synthetic regression and classification on CIFAR datasets are considered. The main claimed findings are focused on 1) the lack of "functional consistency" shown by approximate posteriors (i.e. MFVI or Laplace approximations for BNNs), 2) the issues to sequentially propagate uncertainty in a rigorous way and 3) the ill-calibrated uncertainty of the posterior predictive density models.

**Claims And Evidence:**

The vast majority of claims and ideas introduced are well-supported with evidence from empirical results, technical derivations or well-referenced context. However, I would like to raise some points here that I thought could be imprecise or just odd from my point of view:

- I don't really understand why references on Bayesian inference (for example @ L85) are focused on Robins et al. and other citations of this sort. Weren't the Bayesian ideals mentioned a contribution of other authors and papers produced long before those ones?
- The BMA in Appendix A looks a bit trivial to me, or at least another way of technically rewriting standard Bayesian quantities over probabilistic densities. I am aware of the existence of BMA and its use, but here, it didn't give me the right feeling. I could be wrong on this, but that was what came to my mind when reading it..

**Essential References Not Discussed:**

Idem to prev. review section

**Experimental Designs Or Analyses:**

I do like the way experiments support the main 3 points/parts of the findings, that's great! However, to me the most interesting one of the three parts is for sure the last one from Section 6. To me, that becomes somehow unclear due to the lack of space. The analysis for Fig 6 and 7 is super important, but results are presented in a quick manner with many details poorly discussed...

The way the experiment is designed looks correct to me, but for instance, the intermediary $\pi'$-predictive densities used to sample are also unclear (or I don't have a taste on how to obtain these in a reproducible way), and how the empirical MP is computed in L412 also doesn't give me a good feeling if applied to different BNN models.

**Methods And Evaluation Criteria:**

Similar thoughts as in the previous section on claims and evidence. I do think the paper is on a good direction, exploring a super interesting topic (which, more or less, many in the probabilistic/Bayesian community already suspected for a long long time). The methods are in general well developed, thorough and supported by good evaluation criteria, all mixed with a high scientific spirit imo.

Some additional comments and concerns from the methodology part:
- Why in Section 4, does it come back to the functional posterior predictive perspective? I don't see the point/utility here.
- I am concerned about the assumption of heteroscedastic noise modelling in Sec. 4 being modelled with a $\sigma^{2}_{\theta}(x)$ NN. I think there is more or less a consensus in the probabilistic ML community that this is not a great idea, as it fails to really produce well-calibrated uncertainties out of the regions with high density of datapoints. (See for instance Detlefsen et al. NeurIPS 2019). Therefore, I'm a bit afraid that the use of such networks is somehow deteriorating the correctness of conclusions obtained from the empirical results there..

---
Detlefsen et al. NeurIPS (2019) /  "Reliable training and estimation of variance networks"

**Other Comments Or Suggestions:**

NC

**Other Strengths And Weaknesses:**

NC

**Questions For Authors:**

I do think the paper is in a great direction and it has key values to be worth for acceptance. Due to I still have my doubts and concerns about some parts and points, I do recommend weak acceptance at this point, but I would be also happy to vote for clear acceptance of the manuscript if the rest of the reviewers agree and there are no missing problems that I accidentally ignored.

**Relation To Broader Scientific Literature:**

Apart from my comments made in the first sections, I don't miss anything special here or at least I'm not aware of missing literature/references. The relation and connection with the ideas discussed around in the BNN community are also right to me.

**Theoretical Claims:**

Just one question on the Martingale assumption from Theorem 6.1, used also later in Proposition 6.1. How difficult/easy is satisfying that $\theta_{N}$ is a martingale? I don't really have a feeling here from the NN parameters of different architectures. Could the authors add a bit more on this?

---

> ### Author Rebuttal · Authors · 2025-04-01
>
> Thank you for your careful review and helpful feedback! We are glad you found our paper interesting, and hope that the responses and new results presented below help increase your confidence in backing acceptance of the paper.
> > I don't really understand why references on Bayesian inference […] are focused on Robins et al. and other citations of this sort [...]
>
> Thank you for pointing this out. We agree and will update the references to some of the originals (e.g. [1,2]).
> > The BMA in Appendix A looks a bit trivial to me [...]
>
> We appreciate this appendix indeed presents background that might appear elementary. It introduces terminology and makes our work accessible by detailing common concepts “often underexplained in Bayesian deep learning papers”, quoting reviewer KcGY. We will add a paragraph to better position this appendix. Thanks for highlighting it!
> > Why in Section 4, does it come back to the functional posterior predictive perspective? I don't see the point/utility here.
>
> We think this is worth reiterating here. Pointwise uncertainty quantification (UQ) and BNNs improving UQ over SGD are of course well studied; the novel investigation here is to evaluate BNNs in dynamic settings. Indeed, VI, for example, produces good static results and uncertainty but its functional and uncertainty calibration properties deviate from Bayesian behaviors.
> > I am concerned about the assumption of heteroscedastic noise modelling in Sec. 4 [...]  I'm a bit afraid that the use of such networks is somehow deteriorating the correctness of conclusions obtained from the empirical results there.
>
> Thank you for pointing this out! It is true that the choice of observation model may have a significant impact on the performance and behavior of Bayesian models, and agree this is important to investigate. We highlight that there are relevant ablations already in our appendix (in Appendix F we examine model misspecification and in Appendix D we do a number of ablations on network architecture).
> To show, however, that our empirical findings are not merely an artefact of the difficulty in heteroskedastic noise calibration, __we have repeated our experiment from Figure 3 with homoskedastic noise model__: [see here](https://raw.githubusercontent.com/anon17438/anon-rebuttal-material/main/rebuttal_homoscedastic_figure_3.pdf). We run inference with both a fixed noise variance, and placing an inverse Gamma hyper-prior on it for a fully Bayesian treatment. In both cases, the results agree with the conclusions of Figure 3.
> > How difficult/easy is satisfying that $\theta_N$ is a martingale?
>
> $\theta_N$ is always a martingale under proper Bayesian updating regardless of architecture (subject to some very weak assumptions about the posterior existing). It will typically no longer be a perfect martingale under approximate updates, which is why we are using this as a test for adherence to Bayesian principles. We will add more high level discussion on this to the section to make it clearer.
> >...[Section 6] becomes somehow unclear due to the lack of space. The analysis for Fig 6 and 7 is super important, but results are presented in a quick manner with many details poorly discussed…
>
> Thank you for this important feedback. We do agree the clarity has suffered here from needing to get things in length. We plan to use most of the extra page available for the camera ready to expand on and improve the clarity of this section. In particular, we plan to add more high-level intuition and explanation detail throughout, further interpret the results, and add more insights on the MP behavior itself. We have also repeated the experiments from Figures 7 and 8 with additional dataset, as discussed in the response to Reviewer KcGY. Please let us know if there is anything further you feel we should add.
> > The way the experiment is designed looks correct to me, but for instance, the intermediary pi’-predictive densities used to sample are also unclear (or I don't have a taste on how to obtain these in a reproducible way), and how the empirical MP is computed in L412 also doesn't give me a good feeling if applied to different BNN models.
>
> The $\pi’$ distributions correspond to an initial trained BNN posterior. Our imputation follows the predictive resampling step from the original MP paper. We first draw a random setting of weights from the posterior $\Theta\sim\pi’(\theta)$, then generate a label from the predicted class probabilities $f_\Theta(x^*)$ in the case of classification (but it generalizes). For reproducibility, in our experiments we have taken sequentially increasing seeds. We will make updates to ensure this is clear.
>
> Thank you again for your insightful review which has really helped us to improve the paper.  Let us know if you have any further questions or concerns.
>
> -----------
>
> [1] Jeffreys, H. (1939). _Theory of Probability_. Clarendon Press.
>
> [2] Savage, L. J. (1972). _The foundations of statistics_. Courier Corporation.

---

### Official Review · Reviewer_Tqfi · 2025-03-20

**Overall Recommendation:** 1

**Summary:**

The paper empirically investigates how well popular approximate inference algorithms for BNNs respect the theoretical properties of Bayesian belief updating.

The study tries to examine whether different Bayesian neural network (BNN) posterior approximations adequately capture epistemic uncertainty by analyzing their behavior in both parameter-space i.e. weights and function-space i.e. predictive distributions. Experiments reveal HMC preserves richer functional diversity in high-uncertainty zones, while variational and other approximate methods oversimplify predictions despite training data constraints.

The work is good summary of the insights into the Bayesian inference algorithms. But the studies presented here are broadly understood by the community. The tradeoffs of the different approximate Bayesian inference approaches are well studied.

Lacks novelty.

**Claims And Evidence:**

-

**Essential References Not Discussed:**

-

**Experimental Designs Or Analyses:**

-

**Methods And Evaluation Criteria:**

-

**Other Comments Or Suggestions:**

-

**Other Strengths And Weaknesses:**

-

**Questions For Authors:**

-

**Relation To Broader Scientific Literature:**

These findings are broadly understood. Lacks novelty.

**Theoretical Claims:**

-

---

> ### Author Rebuttal · Authors · 2025-03-31
>
> Thank you for your work in reviewing our paper. However, we believe there may have been some significant misunderstandings about our work and we are a little puzzled by both your summary of our work and the conclusions of your review, neither of which reflect our actual contributions.
>
> For instance, the review claims the paper studies whether “different BNN posterior approximations adequately capture epistemic uncertainty by analyzing their behavior in both parameter-space i.e. weights and function-space i.e. predictive distributions”. However, our work is not about comparing how different posterior approximations capture epistemic uncertainty and we do not have any analysis or experimentation at all about parameter-space behavior.
>
> The tradeoff between various approximate inference techniques in terms of their static predictive performance is indeed already well understood, as we acknowledge in the paper itself.  As explained in, for example, the penultimate paragraph of our introduction, our work is instead about critically examining how well these BNN algorithms adhere to various important theoretical principles of Bayesian belief updating in dynamic settings, which are not in themselves often used to evaluate inference schemes. This goes beyond simply assessing predictive performance or understanding computational and expressivity trade-offs between different approximate inference schemes. The work shifts the focus to systematically examining the behavior of Bayesian updates, looking at functional consistency, coherency of sequential belief updates, and the self-consistency of predictive uncertainty.
>
> Other reviewers recognize this perspective as significant empirical contributions not yet widely established within the community.  We would therefore like to ask you to please reconsider your assessment.

---

### Decision · Program_Chairs · 2025-05-01

**Decision:**

Accept (poster)

**Comment:**

This paper investigates the properties of several popular algorithms for approximate posterior inference in Bayesian neural networks (BNNs). The main experimental findings are that common approximate inference algorithms in BNNs: (a) do not exhibit functional consistency of posteriors, (b) do not propgate information coherently when performing sequential inference, and (c) do not exhibit predictive coherence.  This paper received 3 high-quality reviews, but I have largely ignored the review of Tqfi as it did not provide helpful insights.  Reviewer 8qCC raised some skepticism about reliance on the heteroskedastic noise model, to which the authors provided homoskedastic results (remaining concerns are mostly minor).  Reviewer yeeB is supportive of the paper on balance, but raises concerns about the lack of theoretical analysis.  Reviewer KcGY found the paper "well-argued and engaging" but requested additional experiments, to which the authors provided experiments on the IMDB dataset.  Discounting reviewer Tqfi there is strong consensus that this paper be accepted.